# Single B cell transcriptomics identifies multiple isotypes of broadly neutralizing antibodies against flaviviruses

Jay Lubow[1], Lisa M. Levoir[1¤], Duncan K. Ralph[2], Laura Belmont[1,3], Maya Contreras[1], Catiana H. Cartwright-Acar[1], Caroline Kikawa[1,4,5], Shruthi Kannan[6], Edgar Davidson[6], Veronica Duran[7,8], David E. Rebellon-Sanchez[9], Ana M. Sanz[9], Fernando Rosso[9,10], Benjamin J. Doranz[6], Shirit Einav[7,8,11], Frederick A. Matsen IV[2,4,12,13], Leslie Goo[1]*

1 Vaccine and Infectious Disease Division, Fred Hutchinson Cancer Center, Seattle, Washington, United States of America, 2 Computational Biology Program, Fred Hutchinson Cancer Center, Seattle, Washington, United States of America, 3 Molecular and Cellular Biology Graduate Program, University of Washington, Seattle, Washington, United States of America, 4 Department of Genome Sciences, University of Washington, Seattle, Washington, United States of America, 5 Medical Scientist Training Program, University of Washington, Seattle, Washington, United States of America, 6 Integral Molecular, Inc., Philadelphia, Pennsylvania, United States of America, 7 Division of Infectious Diseases and Geographic Medicine, Department of Medicine, Stanford University School of Medicine, Stanford, California, United States of America, 8 Chan Zuckerberg Biohub, San Francisco, California, United States of America, 9 Clinical Research Center, Fundación Valle del Lili, Cali, Colombia, 10 Department of Internal Medicine, Division of Infectious Diseases, Fundación Valle del Lili, Cali, Colombia, 11 Department of Microbiology and Immunology, Stanford University School of Medicine, Stanford, California, United States of America, 12 Department of Statistics, University of Washington, Seattle, Washington, United States of America, 13 Howard Hughes Medical Institute, Seattle, Washington, United States of America

¤ Current address: Department of Biostatistics, Vanderbilt University Medical Center, Nashville, Tennessee, United States of America

* lgoo@fredhutch.org

**Data Availability Statement:** All relevant data are within the manuscript and its Supporting Information. Full sequencing data that support the

## Abstract

Sequential dengue virus (DENV) infections often generate neutralizing antibodies against all four DENV serotypes and sometimes, Zika virus. Characterizing cross-flavivirus broadly neutralizing antibody (bnAb) responses can inform countermeasures that avoid enhancement of infection associated with non-neutralizing antibodies. Here, we used single cell transcriptomics to mine the bnAb repertoire following repeated DENV infections. We identified several new bnAbs with comparable or superior breadth and potency to known bnAbs, and with distinct recognition determinants. Unlike all known flavivirus bnAbs, which are IgG1, one newly identified cross-flavivirus bnAb (F25.S02) was derived from IgA1. Both IgG1 and IgA1 versions of F25.S02 and known bnAbs displayed neutralizing activity, but only IgG1 enhanced infection in monocytes expressing IgG and IgA Fc receptors. Moreover, IgG-mediated enhancement of infection was inhibited by IgA1 versions of bnAbs. We demonstrate a role for IgA in flavivirus infection and immunity with implications for vaccine and therapeutic strategies.

findings of this study are publicly accessible from: https://doi.org/10.5281/zenodo.7864775.

**Funding:** This work was supported by the Fred Hutchinson Cancer Center Translational Data Science Integrated Research Center New Collaborations Award (LG, FAM, JL, LML, DKR); NIH R01 AI146028 (DKR, FAM); the Howard Hughes Medical Institute (FAM); Viral Pathogenesis and Evolution Training Grant T32 AI083203 (LB); Fred Hutchinson Cancer Center Diverse Trainee Fund (MC); an Investigator Initiated Award W81XWH1910235 from the Department of Defense Office of the Congressionally Directed Medical Research Programs (SE); Catalyst and Transformational Awards from Dr. Ralph & Marian Falk Medical Research Trust (SE); NIH U19 AI057229 supplement (SE); the Chan Zuckerberg Biohub (SE); the Antibody Technology (RRID: SCR_022608), Flow Cytometry (RRID: SCR_022613), and the Genomics & Bioinformatics (RRID:SCR_022606) Shared Resource Facilities of the Fred Hutch/University of Washington/Seattle Children's Cancer Consortium (P30 CA015704); and the Scientific Computing Infrastructure at Fred Hutch (ORIP grant S10OD028685). FAM is an Investigator of the Howard Hughes Medical Institute. SE is a Chan Zuckerberg Biohub - San Francisco Investigator. VD was supported by a Chan Zuckerberg Biohub Collaborative Postdoctoral Fellowship. The funders had no role in study design, data collection and analysis, decision to publish, or preparation of the manuscript.

**Competing interests:** I have read the journal's policy and the authors of this manuscript have the following competing interests: LG, JL, FAM, DKR, LML are inventors on a patent application filed by Fred Hutchinson Cancer Center relating to newly discovered antibodies described in this paper. SK, ED, BJD are employees of Integral Molecular, Inc.; BJD is also a shareholder of the company.

## Author summary

A central challenge for developing clinical interventions for dengue virus or the closely related Zika virus is the ability of IgG antibodies to enhance, rather than neutralize infection under certain conditions. When present prior to infection, as in the case of vaccination, these antibodies can worsen disease outcome. In this study, we analyzed B cells of individuals who experienced dengue or Zika infection to identify those expressing antibodies that can potently neutralize these viruses with minimal potential to enhance infection. We used a method that captured a larger number and wider variety of antibodies than previous approaches. We discovered several potent antibodies that simultaneously neutralized dengue and Zika viruses, including those of IgG isotype, which are common, and one of IgA isotype, which had never been described against this group of viruses. Although IgG antibodies enhanced infection in certain cases, the IgA antibody did not. We further showed that modifying a region of IgG antibodies to convert them to IgA antibodies eliminated their ability to enhance infection. Moreover, the modified IgA versions inhibited the ability of IgG versions to enhance infection. These results suggest that inducing IgA antibodies may be an attractive goal for safe and effective vaccines.

## Introduction

Zika virus (ZIKV) and the four circulating serotypes of dengue virus (DENV1-4) are mosquito-borne flaviviruses with overlapping geographic distributions [1]. Climate change is predicted to further expand the geographic range of mosquito vectors [2–4], highlighting the need for effective clinical interventions to curb epidemics. The complex antibody response to DENV1-4 has hampered the development of safe and effective vaccines. A first exposure to a given DENV serotype generates potently neutralizing antibodies that typically provide long-term, though sometimes incomplete protection against reinfection by that serotype [5–7]. However, antibodies that are cross-reactive in binding but not neutralizing activity against other DENV serotypes are also elicited [8–11] and pre-existing non-neutralizing antibodies predict the risk of severe disease following secondary exposure to a different DENV serotype [12–16]. This phenomenon is attributed to a process called antibody-dependent enhancement (ADE), in which non-neutralizing IgG antibodies [12,17] facilitate the uptake of bound DENV particles into relevant myeloid target cells via Fc-Fc gamma receptor (FcγR)-dependent pathways [18]. ADE-related safety concerns derailed the widespread use of the first licensed DENV vaccine, which increased the risk of severe dengue disease following subsequent infection in previously DENV-naive recipients [19,20]. As pre-existing IgG antibodies from one prior exposure to ZIKV can also enhance subsequent dengue disease risk [21], a safe vaccine would ideally induce durable antibodies that can broadly and potently neutralize DENV1-4 and ZIKV.

In contrast to primary DENV exposure, secondary exposure to a different DENV serotype typically elicits broadly neutralizing antibody responses associated with protection against subsequent disease [8,21–26]. Studying the antibody repertoire in individuals who have experienced multiple DENV infections can thus provide insight into the properties of cross-reactive neutralizing antibody responses that an effective vaccine seeks to mimic. A handful of monoclonal broadly neutralizing antibodies (bnAbs) that can potently neutralize DENV1-4 and in some cases, ZIKV, have been isolated from naturally infected individuals living in endemic regions [22,27–29]. The most well-characterized class of flavivirus bnAbs targets a quaternary E-dimer epitope (EDE) spanning both E protein monomers within the dimer subunit [28,30]. There are two subclasses of EDE bnAbs, of which EDE1 but not EDE2 antibodies can potently

neutralize ZIKV in addition to DENV1-4 [31]. A few antibodies that can cross-neutralize
ZIKV and some DENV serotypes have also been described [32–35], but other than those of the
EDE1 subclass, SIgN-3C is the only known naturally occurring antibody that can potently neu-
tralize ZIKV and all four DENV serotypes [27,36,37].

The above antibodies were discovered by sorting hundreds of single B cells from individuals
infected with DENV and/or ZIKV, followed by either immortalization or PCR amplification
of variable heavy and light chain genes for recombinant IgG production and characterization
[38]. Although these approaches have successfully identified bnAbs against many viruses, they
are laborious, typically requiring robots and/or large teams to increase throughput. As an alter-
native high-throughput method, we previously provided proof-of-principle for a single cell
RNA sequencing (scRNAseq)-based approach to identify multiple DENV1-4 bnAbs, of which
two somatic IgG variants, J8 and J9, were the most potent [39]. Single cell transcriptomics also
allows unbiased profiling of multiple antibody isotypes unlike previous methods, which were
largely restricted to isolation of IgG antibodies [28,33–35,40].

Here, we have improved upon our scRNAseq-based method to discover new bnAbs by sys-
tematically profiling the antibody response in 4 individuals whose sera potently cross-neutral-
ized DENV1-4 and ZIKV. We identified 23 new bnAbs, of which a subset displayed
neutralization breadth and potency comparable or superior to leading bnAbs in the field but
with distinct epitopes. Moreover, one of our newly identified bnAbs neutralized DENV1-4
and ZIKV and is derived from the IgA1 isotype, thus representing the first non-IgG bnAb
described against flaviviruses. Notably, monomeric IgA1 versions of newly and previously
characterized bnAbs not only retained IgG neutralization capacity, but also inhibited IgG-
mediated enhancement of infection in cells expressing both IgG and IgA Fc receptors.

## Results

### Identifying donors with broadly neutralizing antibody responses

We previously identified bnAbs against DENV1-4 [39] via secondary analyses of relatively few
(~350) B cells from an existing scRNAseq dataset of bulk peripheral blood mononuclear cells
(PBMCs). This dataset was generated in an unrelated study with the primary goal of identify-
ing biomarkers of severe dengue [41] in a cohort of individuals with acute DENV or ZIKV
infection [42,43]. Here, we initiated a new study to specifically leverage scRNAseq for bnAb
discovery by focusing our analysis on B cells (instead of bulk PBMCs) from individuals whose
serum broadly neutralized DENV1-4 and ZIKV (**Fig 1**). [42,43]. To identify such individuals,
we screened longitudinal serum samples from 38 cohort participants for their ability to neu-
tralize commonly used DENV1-4 and ZIKV strains in two independent experiments. **S1 Fig**
summarizes the serum neutralization profile of cohort participants, along with demographic
and clinical information. When tested at a single dilution, no serum sample reproducibly neu-
tralized West Nile virus (WNV), a more distantly related flavivirus included as a control. In
contrast, even at the earliest available time point (range: 0 to 7 days after fever onset), serum
samples from 26/38 individuals inhibited infection by two or more DENV serotypes by >50%
in both experiments (**S1 Fig**). This high prevalence of cross-serotype neutralizing activity likely
reflects repeated DENV exposures, as confirmed by IgG avidity testing of these samples in
prior studies [42,43]. In addition to broad neutralizing activity against DENV1-4, serum sam-
ples from 11/38 individuals reproducibly neutralized >50% infection by ZIKV.

### Mining broadly neutralizing antibody repertoires at the single B cell level

To discover monoclonal bnAbs, we chose 4 individuals with cross-flavivirus serum neutraliz-
ing activity, as confirmed by dose-response neutralization assays (**Fig 1A**). In addition to

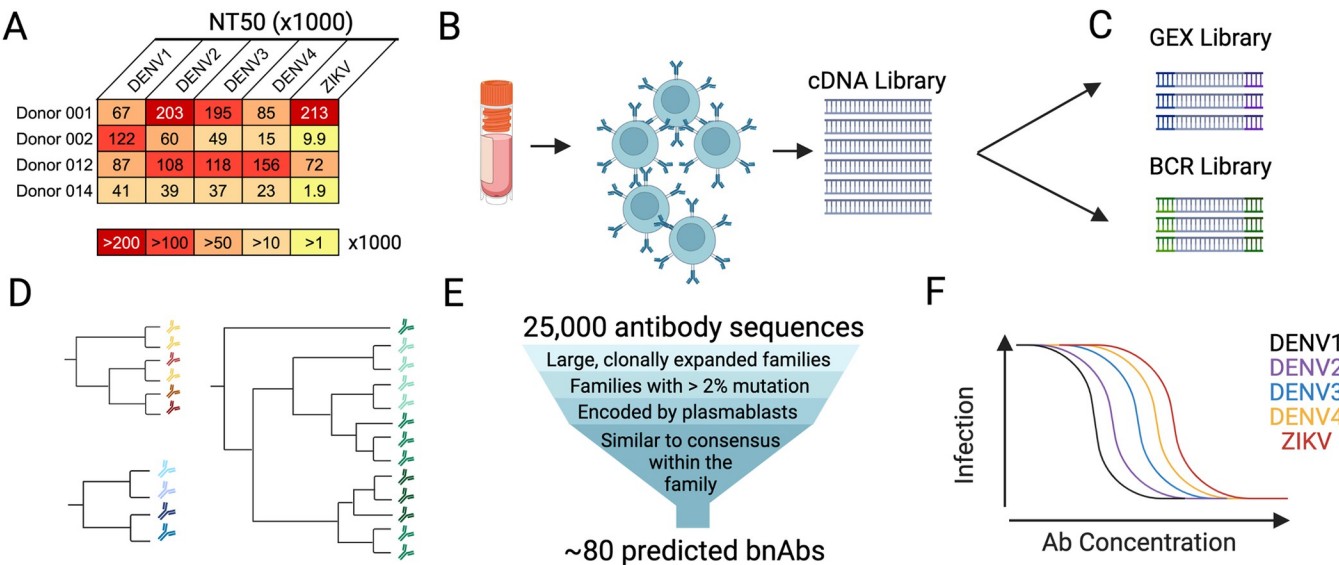

**Fig 1. Workflow to identify broadly neutralizing antibodies (bnAbs) from donor samples. (A)** Serum neutralization profile of 4 cohort participants chosen for downstream analysis based on potent neutralizing activity against DENV1-4 and ZIKV. The mean reciprocal serum dilution that neutralized 50% of virus infection (NT50) in 3 independent experiments is depicted as a heatmap with a darker color indicating greater potency according to the key. **(B)** B cells isolated from the peripheral blood mononuclear cells (PBMCs) of donors selected in (A) were processed for **(C)** single-cell RNA sequencing of both global gene expression (GEX) and B cell receptor (BCR)-specific libraries. **(D)** We analyzed BCR libraries using the software package *partis* [44], which groups antibodies into clonal families and infers their shared ancestry. **(E)** Antibody sequences most likely to encode broadly neutralizing antibodies (bnAbs) were bioinformatically downselected for functional characterization. **(F)** We recombinantly expressed selected antibodies as IgG1 and screened them for the ability to neutralize DENV1-4 and ZIKV. This figure was created with Biorender.com.

serum neutralization breadth and potency, these individuals were selected due to the availability of corresponding PBMCs at early time points during which bnAb responses were detected (within 11 days post-fever onset) (**S1 Fig**). We chose early time points to maximize our likelihood of detecting transiently circulating plasmablasts. This B cell subset undergoes a large expansion following acute DENV exposure [25,40,45–48] and often encodes neutralizing antibodies against multiple DENV serotypes and in some cases, ZIKV, after repeated exposures [25,27,28,39]. Moreover, unlike memory B cells, plasmablasts constitutively secrete antibodies so their antibody repertoire likely mimics that of contemporaneous serum.

We isolated CD19+ B cells from PBMCs of these 4 donors (**Fig 1B**) for scRNAseq of B cell receptor-specific and overall gene expression libraries (**Fig 1C**). We obtained a total of 25,293 paired heavy and light chain antibody coding sequences, with a mean of 6,323 per donor (range 4,644–9,249), comparable to previous studies that profiled antibody repertoires using this method [49–51]. To mine this rich repertoire for flavivirus bnAbs, we first grouped antibodies into clonally related sequences derived from the same rearrangement event (i.e. clonal families, **Fig 1D**) using *partis* [44]. We also used gene expression libraries to determine the B cell subset —naïve, memory, or plasmablast—from which antibody sequences were derived. These analyses allowed us to apply a set of criteria that we and others have found to predict antibody affinity and/or neutralizing activity (summarized in **Fig 1E** and detailed in Methods) to downselect antibodies for neutralization screens (**Fig 1F**). Briefly, we chose antibodies that were 1) from large clonal families with >2% somatic hypermutation, suggesting antigen-specific selection [39,51,52]; 2) encoded by plasmablasts as these are often broadly neutralizing [25,27,28,39]; and 3) most similar to their family's amino acid consensus sequence, suggesting high affinity [53].

As shown in **Fig 2A**, the sizes of clonal families and the distributions of B cell subsets within these samples varied substantially. Samples from donors 001 and 012 were dominated by naïve

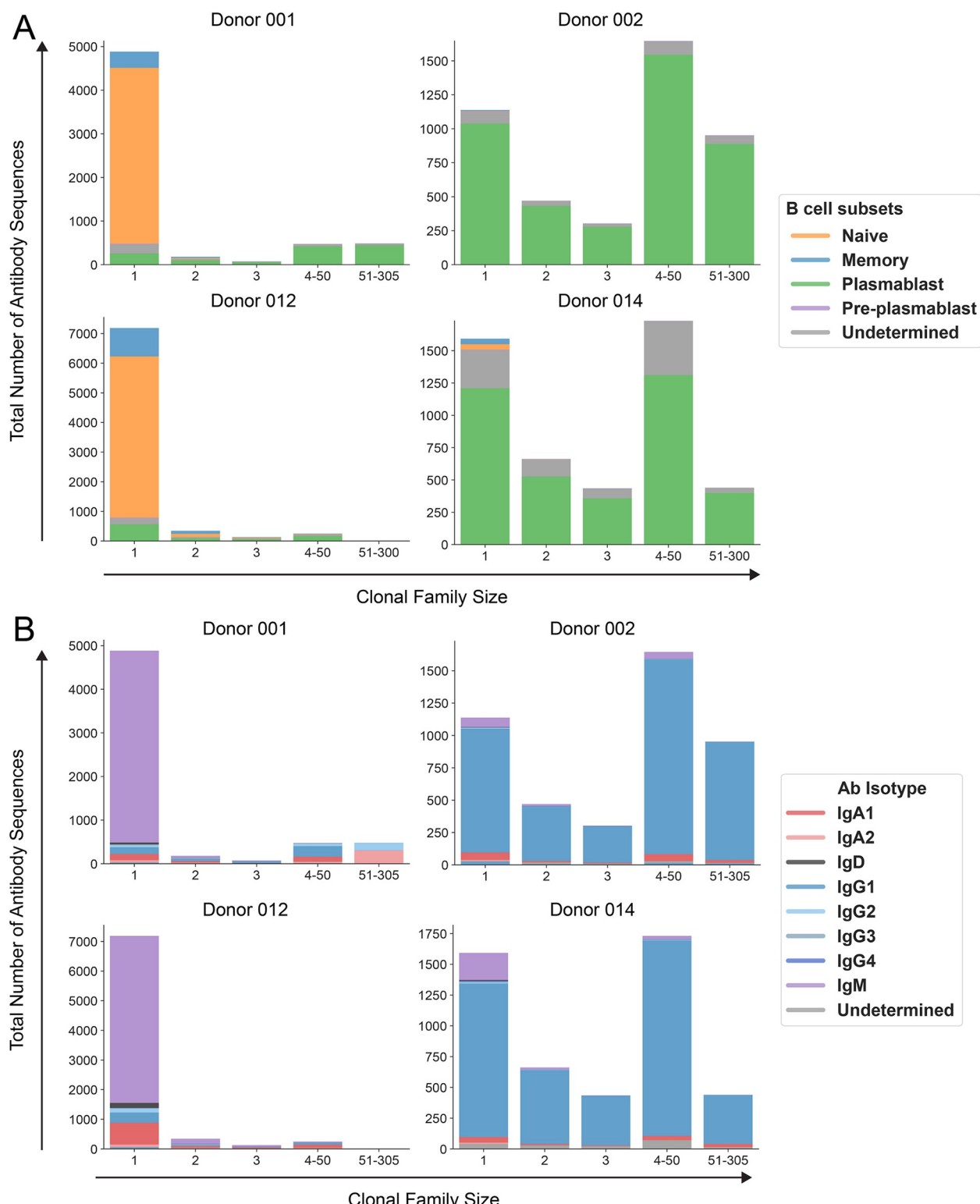

**Fig 2. Distribution of B cell subsets and antibody isotypes within clonal families.** Graphs depict the number of antibodies encoded **(A)** by distinct B cell subsets and **(B)** as various isotypes in clonal families of different sizes in each of the four donor samples analyzed. B cell subset and antibody isotype were determined by analysis of the cell's transcriptome as captured by the gene expression library (see Methods and S4 Table for details). Only B cells for which a corresponding antibody sequence was observed in the B cell receptor library were included. "Undetermined" B cell subset indicates that the cell had too few reads or unique molecular identifiers to yield accurate gene expression information as analyzed by 10X

Genomics Cell Ranger. "Undetermined" isotype indicates insufficient sequence coverage to determine the constant gene segment within the antibody.

B cells that were not members of any clonal family we could discern. By contrast, samples from donors 002 and 014 were composed mostly of plasmablasts, including those in large (4–50 members) or very large (50+ members) clonal families. While most previous methods for antibody discovery have specifically enriched for IgG+ B cells [28,33–35], our scRNAseq-based approach was isotype-agnostic and recorded the native isotype of every sequenced antibody. Like the distributions of B cell subsets, we found that antibody isotype distribution varied by donor: antibodies in samples from donors 001 and 012 were mostly IgM while those from donors 002 and 014 were primarily IgG1 (Fig 2B). The isotype data were not used in our selection algorithm (Fig 1E), though we explored the role of isotype in antibody function, as described later.

## Functional screens for broadly neutralizing antibodies

We performed screening in two rounds. In the first round, our goal was to identify clonal families encoding bnAbs. To do this, we selected 1–3 antibodies from each of ~20 clonal families per donor according to the above criteria (Fig 1E). These antibodies were recombinantly expressed initially as IgG1 by transfection of mammalian cells and the antibody-containing supernatant screened at a single dilution (1:10) for neutralization of DENV1-4, ZIKV, and West Nile Virus (WNV). As controls, we produced and screened previously published antibodies, EDE1-C10 [28,31] and CR4354 [54] in parallel. Consistent with their known specificities, EDE1-C10 broadly neutralized DENV1-4 and ZIKV, but not WNV, while CR4354 specifically neutralized WNV (S2A Fig). Although our downselected antibodies had little to no neutralizing activity (<50%) against WNV, several potently neutralized DENV and/or ZIKV (S2B–S2G Fig; antibodies screened in this first round are left aligned). The number and neutralization profile of clonal families encoding neutralizing antibodies against DENV and/or ZIKV varied by donor. For example, of 14 total families tested from donor 001 (S2B Fig), only two (F05, F07) encoded neutralizing antibodies: F05 displayed ZIKV-specific neutralization, while F07 neutralized DENV1-3 and ZIKV, but not DENV4. Similarly, of the 18 selected families from donor 012 only two (F12, F15) encoded neutralizing antibodies (S2C Fig). In contrast, almost all 25 families from donor 002 neutralized DENV1 and DENV3, and one (F09) broadly neutralized DENV1-4 and ZIKV (S2D Fig). Donor 014 antibodies displayed the broadest neutralization profile (S2E–S2G Fig): almost all 27 selected clonal families neutralized multiple serotypes of DENV and, in some cases, ZIKV with varying potencies. Of these, antibodies from two families (F05 and F09) neutralized DENV1-4 by a mean of 97% and one family (F25) neutralized DENV1-4 and ZIKV by a mean of 92%.

Having identified clonal families encoding bnAbs (bolded in S2B–S2G Fig), we initiated a second round of screening to identify additional bnAbs within those families. Antibodies screened in round two are italicized and indented in S2B–S2G Fig. In general, antibodies within a given family displayed similar neutralization breadth. For example, all 10 antibodies selected from family F07 of donor 001 neutralized DENV1, DENV2, DENV3, and ZIKV, but not DENV4 (S2B Fig). Similarly, all tested antibodies from donor 014 family F09 neutralized DENV1-4 but not ZIKV (S2F Fig), while 6/8 antibodies from family F25 broadly neutralized DENV1-4 and ZIKV (S2G Fig). These results demonstrate that our bioinformatics-based approach successfully identified clonal families encoding multiple bnAbs.

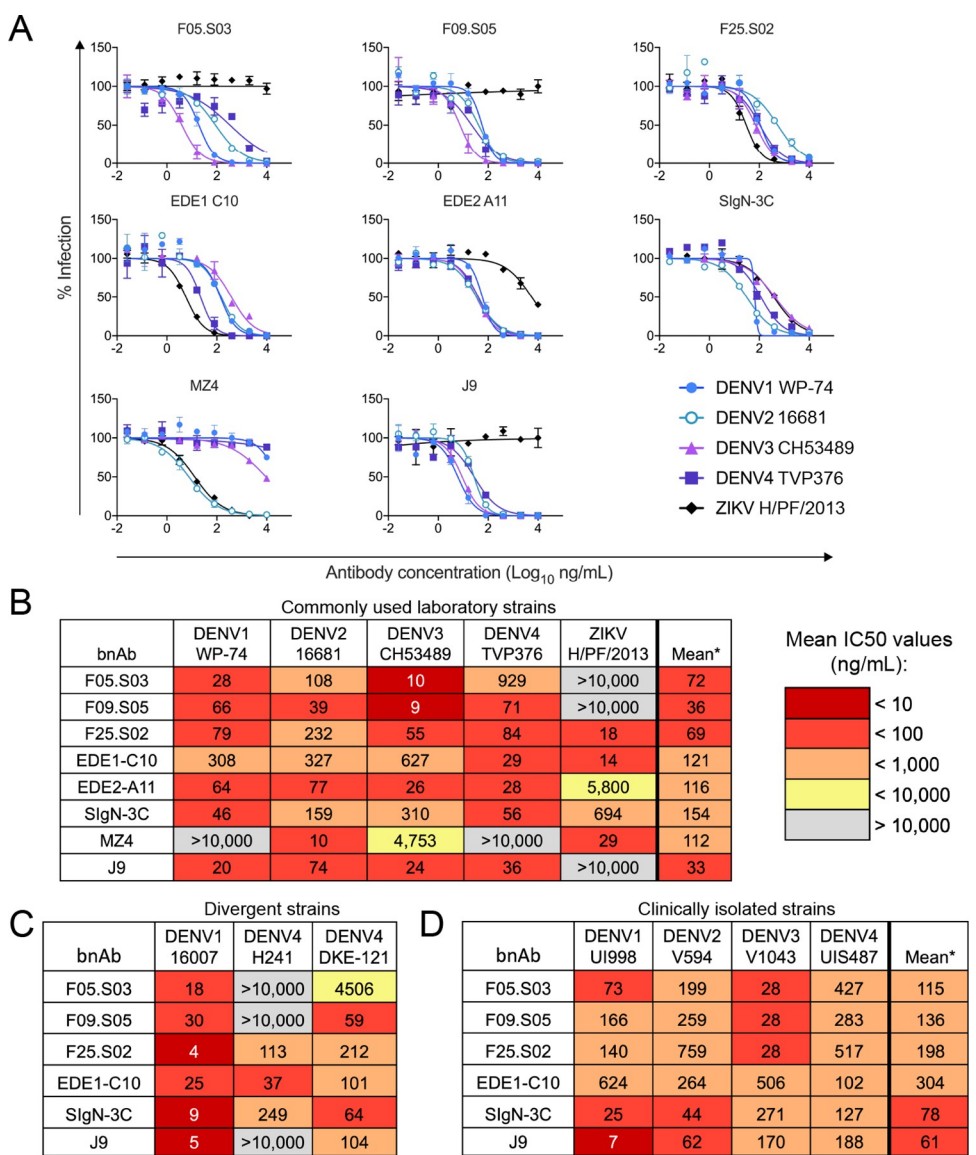

**Fig 3. Neutralization profile of top bnAbs expressed as IgG1. (A)** Representative dose-response neutralization curves of each antibody against DENV1 WP-74, DENV2 16681, DENV3 CH53489, DENV4 TVP376, and ZIKV H/PF/2013 reporter virus particles performed in at least 3 biological replicates in duplicate wells. The data points and error bars represent the mean and range of the duplicates, respectively. **(B)** Mean IC50 values for antibody-virus pairs shown in (A) and compiled from S1 Table. *The final column displays the geometric mean IC50 values against neutralized viruses. **(C)** IC50 values against additional DENV variants selected due to known antigenic divergence from the panel in (B). Values shown are means from at least two biological replicates. **(D)** Mean IC50 values against fully infectious DENV clinical isolates from 2004–2007. Values were obtained from at least two biological replicates. *The final column displays the geometric mean IC50 of each antibody against the four viruses. In (B-D), IC50 values are displayed as heatmaps according to the key. Gray indicates that 50% neutralization was not observed at the highest antibody concentration tested (10,000 ng/ml).

## Functional characterization of the broadest neutralizing antibodies

Based on the above crude screens performed with transfection supernatant (S2 Fig), we purified 23 IgG1 antibodies that inhibited DENV1-4 and in some cases ZIKV by >50% for further characterization. All but one (F15.S01 from donor 012) of these antibodies were from donor 014. We confirmed their neutralizing activities in dose-response assays and calculated the

concentration at which they inhibited 50% of virus infection (IC50) (**S1 Table**). As comparison, we expressed and tested previously identified bnAbs in parallel. These include: EDE1 [28,31] and SIgN-3C [27,37] antibodies, all of which potently neutralize DENV1-4 and ZIKV; EDE2 antibodies, which are distinguished from EDE1 by their weak potency against ZIKV [31]; MZ4, which neutralizes ZIKV and some DENV serotypes [33]; and J9, an antibody we previously isolated from a different donor in the same cohort, which potently neutralizes DENV1-4, but not ZIKV [39].

We assigned antibodies into two categories based on neutralization breadth: 1) those that neutralized DENV1-4 and ZIKV, and 2) those that neutralized DENV1-4 but not ZIKV. Antibodies in each category were ranked based on geometric mean IC50 (**S1 Table**). Among all category 1 antibodies tested, the top-ranking was F25.S02 from donor 014 (geometric mean IC50 value of 69 ng/mL). Compared to previously published category 1 bnAbs, the potency of F25.S02 against ZIKV was similar to EDE1-C10 (IC50 of 18 and 14 ng/ml, respectively) but was ~39 times higher than that of SIgN-3C (IC50 of 694 ng/ml). The geometric mean potency of F25.S02 against DENV1-4 was also ~2-fold higher than that of EDE1-C10 (IC50 of 96 ng/ml versus 207 ng/ml, respectively). Family F25 contained 3 other antibodies that broadly neutralized DENV1-4 and ZIKV. These antibodies (F25.S03, F25.S04, F25.S06) neutralized DENV1, DENV2, DENV3, and ZIKV with relatively similar potency as F25.S02, but they were less potent against DENV4 (IC50 > 1 μg/ml). Among newly identified category 2 antibodies, F09.S05 was most potent; its geometric mean IC50 against DENV1-4 was comparable to the previously identified J9 [39] (36 ng/ml and 33 ng/ml, respectively). Additional high-ranking category 2 antibodies include others from family F09 and antibody F05.S03 from family F05.

Even within the same donor, bnAbs were derived from multiple germline genes and did not display unusually high levels of somatic hypermutation (**S2 Table**), as has been reported for some bnAbs against other viruses [55, 56]. For subsequent detailed characterization, we chose the top-ranking antibody from each clonal family of donor 014, namely F25.S02, F09.S05, and F05.S03. **Fig 3A** shows representative dose-response neutralization assays demonstrating that these new bnAbs were roughly as potent as, and in some cases, more potent than previously published bnAbs (**Fig 3B** and **S1 Table**).

## Newly identified antibodies neutralize flavivirus antigenic variants

There is antigenic variation even within a given DENV serotype [57–59], which is composed of distinct genotypes [60,61]. For example, the DENV1 strain West Pac-74 (WP-74) used in the above screens belongs to genotype IV, which is the most antigenically distinct within this serotype [58]. Additionally, this DENV1 strain is thought to display altered structural dynamics that globally affect antigenicity [62,63]. To rule out the possibility that DENV1 inhibition we observed was limited to an unusually neutralization-sensitive strain, we confirmed that our novel bnAbs also potently neutralized the genotype II DENV1 strain 16007 (IC50 range of 4 to 30 ng/ml, **Fig 3C**). DENV4 also displays antigenic variation across genotypes (I and II) that circulate in humans [64,65]. Many of our newly identified lower-ranking antibodies neutralized the DENV4 genotype II TVP376 strain used in the above screens with modest potency (**S1 Table**). When tested against the DENV4 genotype I strain H241, we found that category 1, but not category 2 bnAbs retained neutralization potency (**Fig 3C**). This preferential neutralization of DENV4 genotype II by most antibodies is consistent with previous observations [65–68]. Among our top-ranking newly identified bnAbs, F25.S02 and F09.S05 neutralized DKE-121 (IC50 of 212 and 59 ng/ml, respectively), a recently described, highly divergent DENV4 strain [69–71], though F05.S03's neutralization of this strain was relatively weak (IC50 of 4500 ng/ml) (**Fig 3C**).

Except for DKE-121, most strains used above were lab-adapted and isolated many decades ago (1956–1982). Additionally, most were tested as single-round infectious reporter virus particles (except for DENV4 H241, which was tested as a replication competent virus). Reassuringly, F25.S02, F09.S05, and F05.S03 also neutralized more contemporaneous, fully infectious DENV1-4 clinical isolates collected between 2004 and 2007 with geometric mean IC50 values lower than for the known bnAb EDE1-C10 but higher than SIgN-3C and J9 (**Fig 3D**).

Aside from genetic diversity, flavivirus antigenic variation can also arise from heterogeneous virion maturation states resulting from inefficient cleavage of prM, a chaperone for the E protein. Many but not all flavivirus-specific antibodies preferentially neutralize incompletely mature virions that retain prM on the surface [72–74]. Importantly, there is increasing evidence that the ability to neutralize the structurally mature form of flaviviruses is important for *in vivo* protection [75,76]. We tested the ability of our novel bnAbs to neutralize partially mature DENV2 or ZIKV produced either under standard conditions or in the presence of excess furin to enhance prM cleavage, resulting in more fully mature viruses [73] (**S3 Fig**). As controls, we included antibodies E60 and ZV-67, which poorly neutralize mature forms of DENV2 and ZIKV, respectively, resulting in a large fraction of non-neutralized virions even at high antibody concentrations (**S3A Fig**), consistent with previous studies [73,77,78]. In contrast to these control antibodies, F25.S02, F09.S05, and F05.S03 potently neutralized DENV2 regardless of maturation state (maximum IC50 fold change of 2.7) (**S3A and S3B Fig**). Moreover, F25.S02 was *more* potent against the mature form of ZIKV (15-fold decrease in IC50) (**S3B Fig**). We also observed preferential neutralization of mature ZIKV by known bnAbs EDE1-C10 and SIgN-3C (**S3A and S3B Fig**).

Overall, these results demonstrate that our new bnAbs can neutralize flavivirus antigenic variants arising from both genetic and structural heterogeneity that are relevant for vaccine efficacy [67,68,76], though the ability to broadly neutralize multiple DENV4 genotypes was restricted to F25.S02.

## Mapping E protein determinants of antibody binding

Many potently neutralizing flavivirus antibodies target complex epitopes displayed optimally on virions and not on soluble monomeric E protein [79]. To determine the E protein oligomeric form recognized by our bnAbs, we performed ELISA to assess binding to soluble monomeric E protein or to virus particles of the prototype DENV2 16681 strain. Unlike antibody B10, which we previously showed to efficiently bind E proteins displayed in both contexts [39], F25.S02, F09.S05, and F05.S03 bound efficiently to E proteins displayed on virus particles only, similar to the known bnAb EDE1-C10 [28] (**Fig 4A and 4B**). These results suggest that our newly identified bnAbs preferentially recognize quaternary epitopes.

To identify E protein amino acid residues critical for binding, we screened antibodies against a shotgun alanine-scanning mutagenesis library of DENV2 prM/E proteins [39,81]. As controls, we included known bnAbs EDE1-C10 and J9. We identified alanine mutations that specifically reduced F25.S02 or F05.S03 binding by >70% relative to wild type DENV2 (**Fig 4C–4F** and **S3 Table** shows screen results against the entire library). Despite testing multiple conditions, we did not detect binding of F09.S05 to wild type DENV2 in this format.

For F25.S02, all E residues identified as important for binding were located in domain II (G78, L82, V97, I113, N242) with the exception of M6 in domain I (**Fig 4C and 4D**). Mutation at these residues minimally impacted binding by the known bnAb EDE1-C10, which retained 50–85% of wild type binding reactivity (**Fig 4D**). EDE1-C10 and F25.S02 are further distinguished by their dependence on K310A, which abolished binding by EDE1-C10, but not by F25.S02 (**Fig 4F**). Thus, although F25.S02 and EDE1-C10 display a similar neutralization profile against DENV1-4 and ZIKV, their binding determinants on DENV2 are distinct.

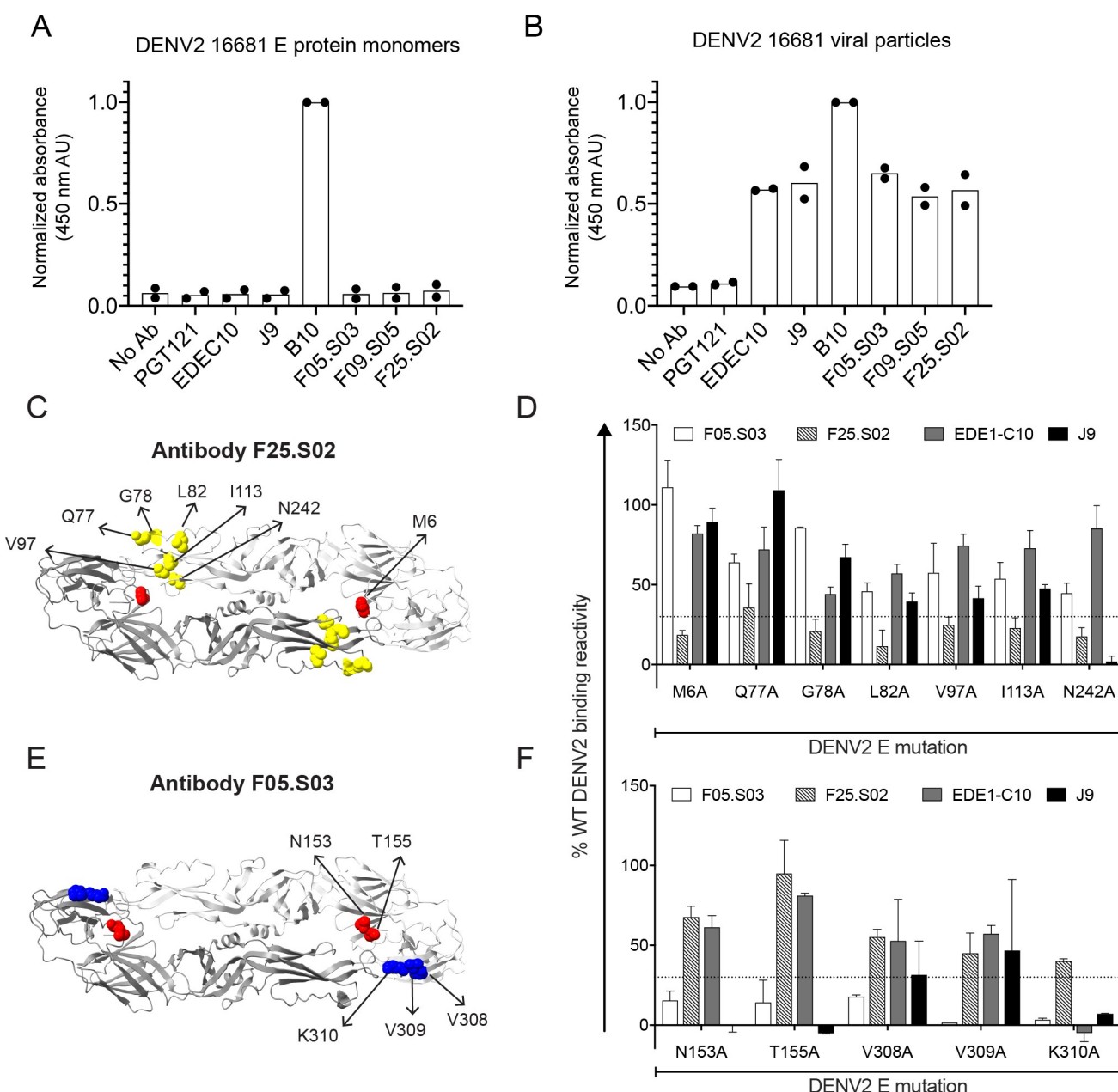

**Fig 4. Determinants of E protein binding by bnAbs.** Relative binding efficiency of the indicated antibodies to **(A)** E protein monomers **(B)** or reporter virus particles of DENV2 16681 measured by ELISA. Results are from two independent experiments, each performed in duplicate wells. The absorbance of each duplicate, reported in arbitrary units (AU), was normalized to the wells that received positive control antibody B10 [39]. The HIV-specific antibody PGT121 [80] was used as a negative control. Data points represent the normalized means of each experiment and the bars represent the means of the two experiments. **(C-F)** DENV2 16681 E protein sites important for binding by antibody **(C)** F25.S02 or **(E)** F05.S03 are shown on the ribbon structure of the DENV2 E dimer (PDB: 1OAN) and labeled on one monomer. Sites in E domains I, II, and III are depicted in red, yellow, and blue, respectively. Bar graphs show the mean binding reactivity to individual alanine mutants that selectively impact **(D)** F25.S02 or **(F)** F05.S03 as a percentage of wildtype (WT) DENV2 E protein reactivity. Binding of control antibodies EDE1-C10 and J9 to these mutants was tested in parallel. Error bars show the range of binding reactivity from two independent experiments. The dotted line in (D-E) indicates 70% reduction in antibody binding activity to mutant compared to WT.

For F05.S03, we identified two clusters of mutations that were important for binding. The first, at E residues N153 and T155 in domain I, abolishes a potential *N*-linked glycosylation site and reduced binding efficiency by ~85% (**Fig 4E and 4F**). The presence of this potential *N*-

linked glycosylation site has also been shown to be important for recognition by J9 [39] (**Fig 4F**) and by the EDE2 subclass of bnAbs [28], all of which potently neutralize DENV1-4, but not ZIKV (**Fig 3**). The second cluster, which is composed of residues V308, V309, and K310, is in domain III but is near the domain I *N*-linked glycosylation site (**Fig 4E**). Of these mutations, K310A also strongly reduced binding efficiency by J9 and EDE1-C10 (**Fig 4F**).

## Mapping neutralization determinants

As F09.S05 neutralized DENV1-4 but not ZIKV, we screened neutralizing activity against a previously described DENV library encoding mutations at solvent accessible E residues that were identical or similar across representative DENV1-4 strains but different from ZIKV [39]. Specifically, amino acids at these E protein sites in DENV2 16681 were substituted with corresponding ZIKV H/PF/2013 amino acids individually or in combination to identify those that reduce antibody potency against DENV2 and thus comprise the neutralization epitope. We also tested a subset of DENV2 alanine mutations identified in the binding screen above to validate their role in neutralization.

Except for the K310A mutation in E domain III, which reduced F09.S05 potency by ~14-fold, mutations that strongly impacted F09.S05 neutralizing activity were in domain I (**Fig 5A**). Removing the potential *N*-linked glycosylation site through mutation at residue N153 or T155 abrogated neutralization, while the nearby V151T mutation reduced F09.S05 potency by ~50-fold. Combining V151T with H149S abolished neutralizing activity. These glycosylation site mutations also abolished neutralization by F05.S03 (**Fig 5B**) and J9 (**Fig 5C**), consistent with results from our binding screen above (**Fig 4F**) and our previous study with J9 [39]. In addition to these shared residues important for neutralization, we identified determinants that distinguished F09.S05 and F05.S03 from each other and from the previously characterized J9. For example, although the individual S145A and H149S mutations minimally impacted F09.S05 (**Fig 5A**) and J9 (**Fig 5C**) (maximum of 5-fold change in IC50), each mutation reduced F05.S03 neutralization potency by ~20-fold (**Fig 5B**). Moreover, the combination of K47T+F279S mutations in domain I minimally impacted F09.S05 and F05.S03 (< 4-fold IC50 change, **Fig 5A and 5B**), but reduced J9 potency by 76-fold (**Fig 5C**).

In contrast to their effects on F09.S05, F05.S03, and J9, mutations at N153 and T155 *increased* neutralization potency of EDE1-C10 and F25.S02 by up to 50-fold (**Fig 5D and 5E**). Another shared feature between EDE1-C10 and F25.S02 is a reduced neutralization potency against the K47T+F279S double mutation in E domain I (36- and 14-fold IC50 increase, respectively, **Fig 5D and 5E**). However, there were distinct neutralization determinants for these bnAbs. Specifically, the I113A and N242A mutations in domain II each reduced F25.S02 potency by ~30-fold (**Fig 5D**) but minimally impacted EDE1-C10 neutralization (<4-fold IC50 change, **Fig 5E**). Conversely, the K310A mutation in domain III strongly reduced EDE1-C10 (~90-fold IC50 increase, **Fig 5E**) but not F25.S02 potency (0.7-fold IC50 change, **Fig 5D**). These results are consistent with the alanine binding screen (**Fig 4D**). Thus, despite some similarities, we identified E residues that differentially impacted neutralization by newly discovered bnAbs relative to each other and to known bnAbs, suggesting they have distinct epitope specificities.

## Effect of antibody valency on neutralizing activity

To gain additional insight into the epitope specificities, we compared the neutralization potency of F25.S02, F09.S05, and F05.S03 tested as bivalent IgG1 or monovalent Fab against DENV2 and ZIKV (**S4 Fig**). Except for F09.S05, the Fab versions of all antibodies tested, including known bnAb controls EDE1-C10 and SIgN-3C, failed to neutralize DENV2 by at

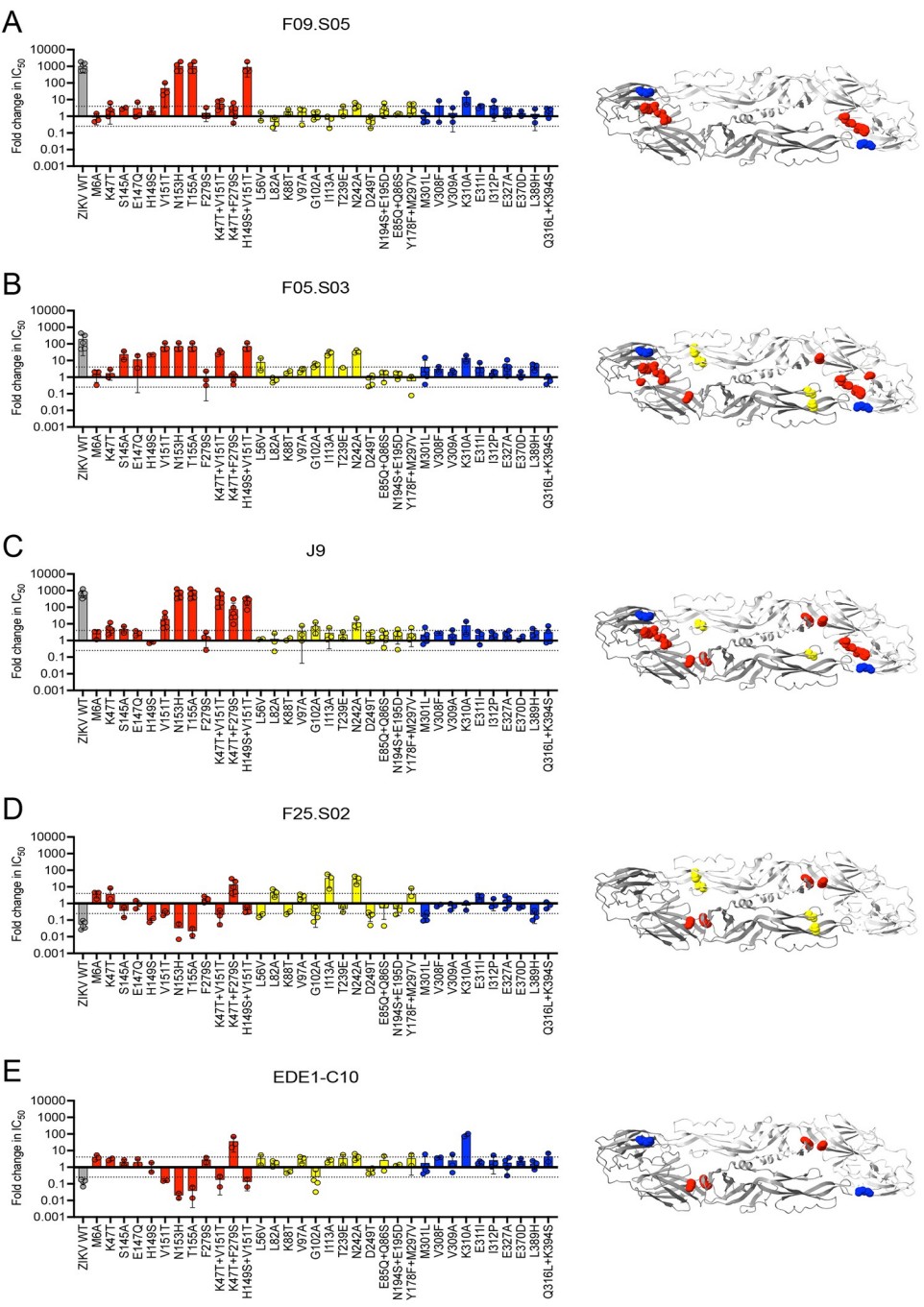

**Fig 5. E protein residues critical for neutralization by bnAbs.** (Left panel) Bar graphs show the mean IC50 fold change against DENV2 16681 reporter virus particles encoding E protein variants relative to wild type (WT) DENV2 for antibodies **(A)** F09.S05, **(B)** F05.S03, **(C)** J9, **(D)** F25.S02, and **(E)** EDE1-C10. Sites in E domains I, II, and III are shown in red, yellow, and blue, respectively. Values of 1, >1, and <1 indicate no change, decreased sensitivity, and increased sensitivity of mutant relative to WT DENV2, respectively. Mean values were obtained from at least 2 independent experiments shown as individual data points in which WT and mutant DENV2 were tested in parallel. WT ZIKV H/PF/2013 (gray) was included as a control. Error bars indicate the standard deviation (n>2) or range (n = 2). In each graph, the dotted horizontal line represents a 4-fold IC50 change. (Right panel) For each bnAb, sites of mutations that reduced neutralization potency when tested either individually or in combination by > 4-fold are depicted as spheres on both monomers of the DENV2 E dimer subunit (PDB 1OAN).

least 50% at the highest antibody concentration tested (400 nM), suggesting that bivalent engagement is important for potent DENV2 neutralization by these antibodies [82]. Although SIgN-3C IgG1 neutralized ZIKV with moderate potency, no neutralization was detected with Fab, consistent with previous findings [37]. In contrast, EDE1-C10 and F25.S02 retained the ability to completely neutralize ZIKV as Fab. Although IgG1 versions of EDE1-C10 and F25.S02 neutralized ZIKV with similar potency, their Fab neutralization profiles were more distinct; unlike EDE1-C10 Fab, which retained relatively potent neutralization consistent with previous findings (<10-fold increase in IC50 compared to IgG) [82], F25.S02 neutralized ZIKV with much reduced potency as Fab (64-fold increase in IC50 compared IgG). These results suggest that EDE1-C10, SIgN-3C, and F25.S02 target distinct epitopes on ZIKV.

## Neutralizing activity of IgA1 antibodies is similar to or better than IgG1 versions

As neutralizing activity is traditionally thought to be dependent mainly on changes within the antibody variable region, neutralizing antibodies have typically been tested as the IgG1 subclass, regardless of their native isotype [83]. Moreover, most studies profiling the neutralizing antibody repertoire against flaviviruses have specifically isolated IgG antibodies [28,33–35,40,45]. While we did not bias our scRNAseq-based approach towards a particular antibody isotype, we initially expressed and screened all antibodies as IgG1, similar to previous studies. Given increasing evidence that antibody Fc isotype can impact neutralizing activity against many viruses [84–89], we used scRNAseq data to confirm that the native isotype of almost all 23 antibodies downselected for detailed characterization was indeed IgG1 (**S2 Table**). However, unlike other flavivirus bnAbs described here or previously, our top-ranking bnAb, F25.S02, was derived from the IgA1 isotype.

To investigate the impact of isotype on neutralizing activity, we expressed F25.S02, EDE1-C10, and SIgN-3C as monomeric or dimeric IgA1 and compared their neutralization profile to IgG1 versions. Although we purified IgA1 dimers by size-exclusion chromatography (SEC), we could not exclude the presence of higher order polymers [90] by SDS-PAGE analysis (**S5 Fig**) so we refer to these antibodies as polymeric IgA1 hereafter.

As shown in **Fig 6**, all 3 bnAbs retained neutralization breadth and potency as monomeric IgA1. Moreover, while F25.S02 monomeric IgA1 and IgG1 displayed comparable potency against DENV1-4 and ZIKV (maximum of 2-fold IC50 change), monomeric IgA1 versions of EDE1-C10 and SigN-3C were more potent against some viruses (**Fig 6B**). For example, compared to their IgG1 versions, EDE1-C10 and SigN-3C monomeric IgA1 antibodies were ~4 times more potent against DENV3, though sample sizes (n = 3) were too small to achieve statistical significance. SigN-3C potency against ZIKV was also 9 times higher as monomeric IgA1 compared to IgG1.

Antibody expression as polymeric IgA1 further increased potency compared to IgG1 to varying extents. This effect was most apparent for viruses against which the IgG1 version of the particular antibody was the least potent; for F25.S02, EDE1-C10 and SigN-3C polymeric IgA1, the largest IC50 reduction compared to IgG1 was observed against DENV2 (20-fold), DENV3 (9-fold), and ZIKV (167-fold), respectively (**Fig 6B**). This increased potency of IgA1 bnAbs is unlikely due to non-specific effects as none neutralized the more antigenically distant WNV (**Fig 6A**).

## IgA1 antibodies inhibit enhancement of infection by IgG1

Virtually all IgG antibodies can enhance flavivirus infection *in vitro* at sub-neutralizing concentrations, presumably by facilitating uptake of IgG-virus complexes into FcγR-expressing

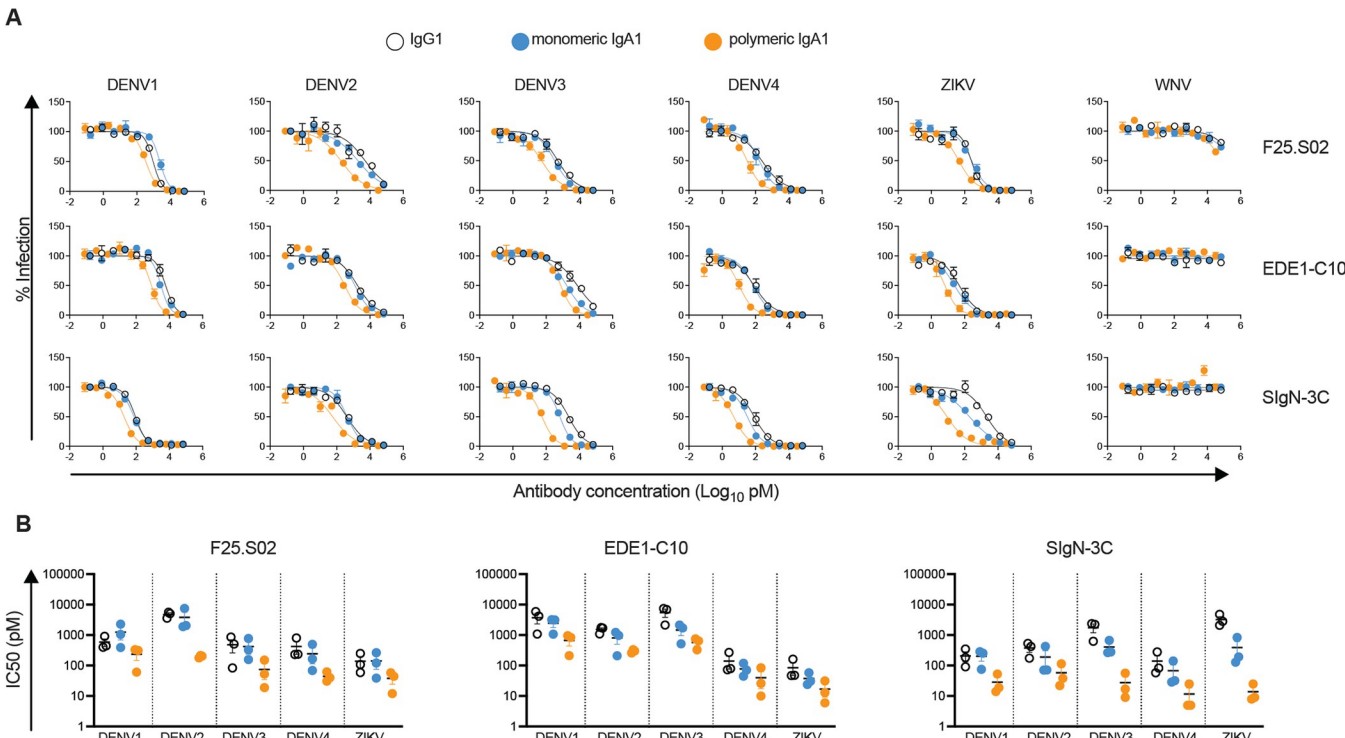

**Fig 6. Neutralization profile of antibodies expressed as IgA1. (A)** IgG1 (open circles), monomeric IgA1 (blue circles), and polymeric IgA1 (orange circles) versions of F25.S02 (top row), EDE1-C10 (middle row), and SigN-3C (bottom row) were tested for their ability to neutralize DENV1 WP-74, DENV2 16681, DENV3 CH53489, DENV4 TVP376, and ZIKV H/PF/2013 reporter virus particles. Dose-response curves are representative of 3 independent experiments, each tested in duplicate wells. Data points and error bars represent the mean and range of the duplicates, respectively. **(B)** Comparison of IC50 values of F25.S02 (left), EDE1-C10 (middle), SigN-3C (right) expressed as IgG1, monomeric IgA1, and polymeric IgA1 against the viruses indicated on the x-axes. Color scheme is similar to (A). Each data point represents an independent experiment in which antibody isotypes were tested in parallel. Horizontal bars indicate the mean. Error bars represent the standard error of the mean.

cells [91]. Accordingly, IgG1 versions of newly and previously identified bnAbs enhanced infection to various extents in K562 cells (S6A Fig) commonly used to study ADE as they express FcγRIIa (S7 Fig) and are poorly permissive to flavivirus infection in the absence of IgG [92]. We did not detect enhancement of ZIKV infection by J9, F09.S05, and F05.S03 (S6A Fig). As IgG-bound, but not naked virions efficiently infect K562 cells, this finding suggests an inability for these antibodies to bind ZIKV, which could explain their lack of ZIKV neutralizing activity (Fig 3A and 3B).

As existing studies of ADE of viral infection or disease have focused on the role of IgG-FcγR interactions [12,17,21,93–95], we next investigated the role of IgA in enhancing DENV infection. We first tested the ability of monomeric IgA1 versions of F25.S02, EDE1-C10, and SIgN-3C to enhance infection of DENV in K562 cells. For these experiments, we chose DENV1 and DENV4 as the infectivity curves obtained across the concentration range of IgG1 versions of bnAbs of interest fully captured both enhancement and neutralization in K562 cells (S6A Fig). As expected, IgG1 but not IgA1 versions of F25.S02, EDE1-C10, and SIgN-3C enhanced DENV infection in K562 cells (S6B Fig), which do not express Fc alpha receptor (FcαRI) (S7 Fig).

We next investigated ADE in U937 monocytic cells, which express FcαRI in addition to FcγRIIa (S7 Fig) [96,97]. For these experiments, we used a concentrated preparation of DENV2 as we observed relatively inefficient IgG1-mediated ADE in U937 compared to K562 cells. Although we obtained the canonical dose-response ADE curve using IgG1 versions of

bnAbs, neither monomeric nor polymeric IgA1 antibodies enhanced DENV infection in U937 monocytes (**Fig 7A**). Moreover, competitive ADE assays using mixtures of IgG1 and monomeric IgA1 antibodies at various ratios demonstrated that autologous IgA1 antibodies inhibited IgG1-mediated ADE of DENV infection in U937 cells (**Fig 7B**) in a dose-dependent manner (**Fig 7C**). This effect was also observed for all three bnAbs in K562 cells (**S6B Fig**), indicating that IgA1 antibodies can interfere with IgG1-mediated ADE in multiple cell types regardless of native isotype and epitope specificity. Crucially, an isotype control IgA1 antibody had virtually no effect on ADE mediated by IgG1 in U937 (**Fig 7B and 7C**) or K562 (**S6B Fig**) cells, indicating that inhibition was due neither to a reduction in IgG1 concentration in IgG1/IgA1 mixtures nor the presence of non-specific IgA1. Rather, IgA1 inhibited ADE mediated by IgG1 likely via direct competition of binding to virions.

## Discussion

Unlike most antibody discovery approaches that involve screening large panels of antibodies expressed by sorted B cells [38], we previously established a proof-of-concept for a bioinformatics-based strategy to identify not only antigen-specific antibodies, as shown previously by other groups [51,98,99], but also those with broadly neutralizing activity [39]. Here, we have improved upon our previous approach and leveraged scRNAseq of B cells to identify multiple antibodies that broadly and potently neutralized DENV1-4 and in some cases, ZIKV. Previous studies characterizing flavivirus bnAb responses have used antibody isolation protocols that specifically enriched the IgG isotype [28,33–35]. In contrast, our scRNAseq approach is designed to capture full-length antibody sequences in an unbiased manner. Although most new bnAbs we discovered were of the IgG1 isotype, consistent with previous findings [27,28,39], we also describe for the first time an IgA1 antibody with broadly neutralizing activity against DENV1-4 and ZIKV.

Despite broad and potent serum neutralizing activity in all 4 donors selected for antibody repertoire analysis, almost all monoclonal bnAbs were isolated from only one donor (014). Although we did not set out to formally investigate the basis for donor-dependent effects, consistent with previous findings [46,47], antibody neutralizing activity could be partly explained by sample collection time (**S1 Fig**), which likely affected our ability to capture transiently circulating plasmablasts (**Fig 2A**), many of which encode bnAbs [25,27,28,39]. Alternatively, the observed serum neutralization breadth and potency across donors could be due to a combination of antibodies with multiple specificities. However, within a given donor, we did not detect an obvious pattern of complementary neutralizing activity among antibodies from distinct clonal families to support this hypothesis (**S2B–S2G Fig**). The number and order of prior flavivirus exposures also impact bnAb development [12]. It is interesting that unlike other donors analyzed, donor 014 was confirmed to have been acutely co-infected with two DENV serotypes (**S1 Fig**). Prior studies have documented concurrent infection by multiple DENV serotypes in hyperendemic regions [100–105], but whether co-infection uniquely impacts bnAb induction has not been systematically explored. Finally, while we successfully identified multiple new bnAbs, our *in silico* down-selection criteria are likely subject to stochastic processes to some extent [106].

Although neutralizing activity is thought to be primarily determined by somatic hypermutation within antibody variable regions, Fc isotype can also impact neutralization potency and/or breadth against many viruses, including flaviviruses [84–88]. For example, a recent study described a naturally occurring ZIKV-specific pentameric IgM antibody (DH1017.IgM) whose potency depended on the IgM isotype [88]. Unlike DH1017.IgM, which did not neutralize DENV, here we identified F25.S02, an IgA1 antibody that potently cross-neutralized ZIKV

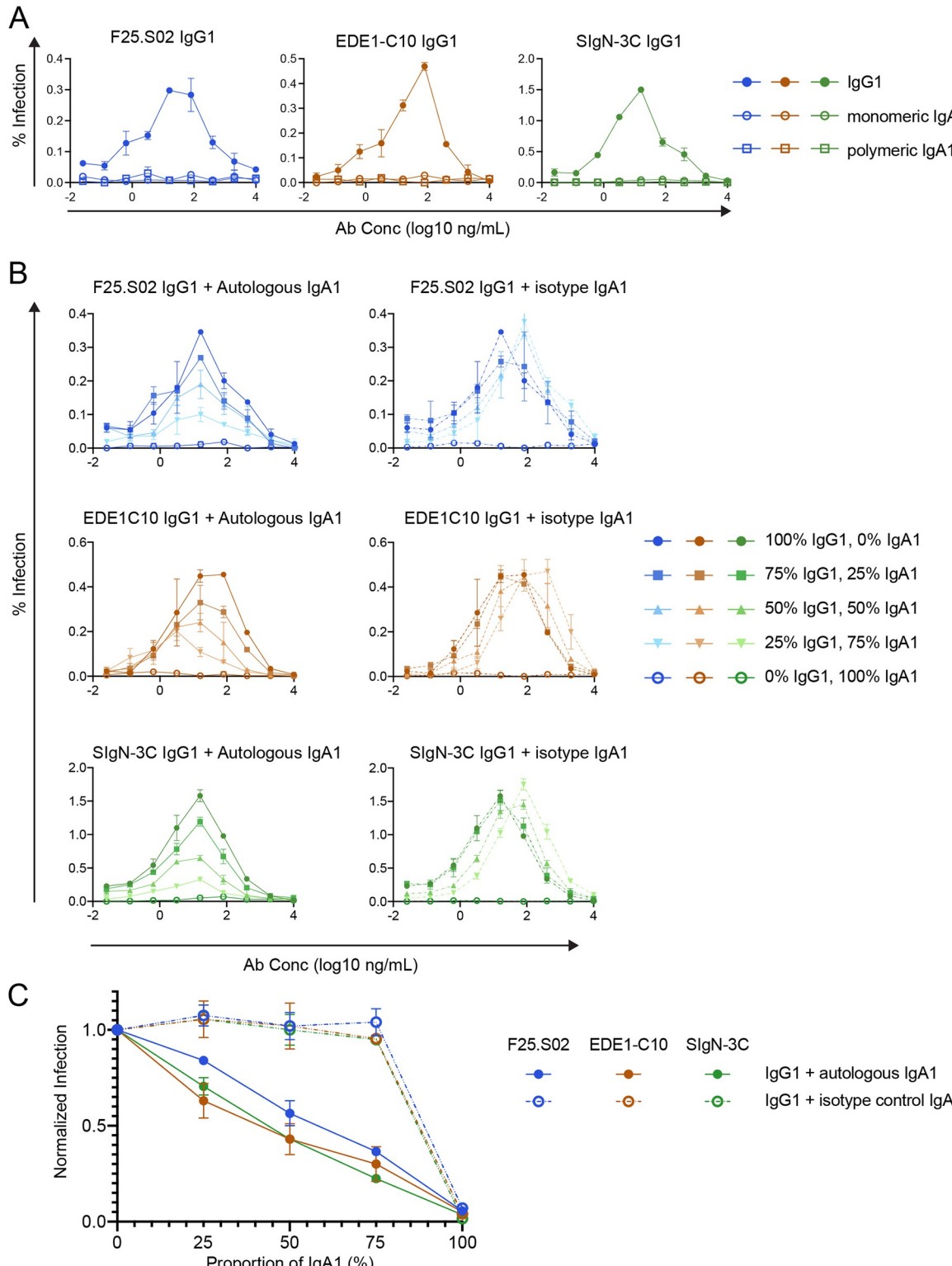

**Fig 7. Effect of antibody isotype on antibody dependent enhancement (ADE) of infection of U937 monocytes. (A)** DENV2 16681 reporter virus particles were pre-incubated with serial dilutions of IgG1 (filled circles), monomeric IgA1 (open circles), or polymeric IgA1 (open squares) forms of F25.S02 (blue), EDE1-C10 (orange), or SIgN-3C (green) prior to infection of U937 cells, which express Fc receptors for both IgG and IgA. IgG1 and IgA1 antibodies were tested individually in the assay. **(B)** Competitive ADE assays in U937 monocytes. F25.S02 (top row, blue), EDE1-C10 (middle row, orange) or SIgN-3C (bottom row, green) IgG1 was mixed with either

autologous IgA1 (left panel, solid lines) or an IgA1 isotype control (right panel, dashed lines) at the indicated ratios by mass before serial dilution and pre-incubation with concentrated DENV2 16681 reporter virus particles. The assay performed in two independent experiments, each in duplicate wells. The data points and the error bars represent the means and the range of the duplicates, respectively, from one representative experiment. **(C)** Area under the curve analysis for experiments represented in (B). For both biological replicates the area of the curve for each infection condition was calculated and normalized to infection in the 100% IgG1 condition. The data points and error bars represent the mean and the range of two independent experiments, respectively.

and DENV1-4 and retained its potency as IgG1. In addition to its distinct isotype, our epitope mapping results demonstrate that despite some similarities, F25.S02 has unique binding and neutralization determinants compared to EDE1-C10 [28,30,31] and SIgN-3C [27,36,37] IgG1 antibodies, which represent the only 2 known classes of bnAbs that potently neutralize DENV1-4 and ZIKV.

While IgA bnAbs have been described for antigenically distinct viruses that infect mucosal surfaces such as HIV [83,107], SARS-CoV-2 [85], and common respiratory viruses [108], to our knowledge, F25.S02 is the first known IgA bnAb against flaviviruses. A recent study showed that infection by the malaria parasite, *Plasmodium falciparum*, another mosquito-borne pathogen, induces serum IgA production in humans that contribute to protection against disease [109]. These results indicate a functional role for IgA even in the context of infections that do not occur primarily at mucosal surfaces.

Human IgA antibodies in serum and mucosal sites exist primarily as monomeric or dimeric/polymeric forms, respectively [90]. As monomeric IgA1, F25.S02 displayed comparable neutralizing activity to IgG1 against DENV1-4 and ZIKV. In contrast, we show that expression of EDE1-C10 and SIgN-3C bnAbs as monomeric IgA1 improved potency against some viruses, despite their native IgG1 isotype [28,40]. These findings are consistent with epitope- and virus-dependent effects of antibody isotype on neutralization [83]. Expression of all 3 bnAbs as polymeric IgA1 increased potency relative to corresponding monomeric IgA1 or IgG1 versions to varying extents, depending on the virus/antibody combination. Defining the mechanism(s) behind this observation awaits further studies but it suggests that the epitope arrangement of these bnAbs allows multivalent engagement by polymeric IgA on the same virion in a context-dependent manner. Alternatively, or in addition to this mechanism, polymeric IgA could bind the same epitope on multiple virions to cause aggregation. Both mechanisms of virion engagement have been shown for DH1017.IgM, depending on the particular antibody conformation [88].

Compared to other isotypes, IgA1 antibodies have a greater distance between Fabs relative to each other and to the Fc domain [110,111], providing a possible mechanism for unique neutralizing and Fc-dependent effector functions [83]. Further, engagement of IgA with FcαRI is distinguished from engagement of other isotypes with their Fc receptors in terms of stoichiometry, orientation, and location of protein binding sites [112], which could impact the efficiency with which different antibody isotypes facilitate ADE. These differences may explain our observation that IgG1, but not IgA1 antibodies mediated ADE of DENV in U937 cells (**Fig 7A**). The significance of this finding is highlighted by another group's almost concurrent observation that the IgG1 but not IgA1 form of a DENV-specific, neutralizing antibody mediated ADE in primary monocyte-derived macrophages, and that FcαRI expression on circulating monocytes was limited during acute DENV infection [113]. Here, we further show that IgA1 versions of bnAbs inhibited IgG1-mediated ADE in a dose-dependent manner, likely via competition for binding to virions. Combined, these results indicate that IgA1 forms of antibodies may offer protection against DENV with minimal risk of enhancement of infection.

Existing studies of flavivirus immunity have heavily focused on the role of IgG antibodies and their interactions with FcγRs [12,16,17,21,93,114]. Although the *in vivo* relevance of our

results remains to be validated, they nevertheless highlight an underappreciated role for flavivirus-specific IgA antibodies in infection and immunity. Indeed, recent studies reported a high proportion of DENV-reactive IgA-expressing plasmablasts following acute primary infection and to a lesser extent, secondary infection [49,115,116]. Our analysis of circulating B cell repertoires here also demonstrates that while IgG dominated the response, IgA and IgM antibodies were prevalent (**Fig 2B**). Notably, FcαRI is expressed on myeloid cells, including monocytes, macrophages, and dendritic cell subsets [117–120], all of which also express FcγRs and are thought to be principal target cells for DENV *in vivo* [41,121–125]. Intriguingly, IgA-FcαRI interactions can modulate activating or inhibitory responses mediated by *other* Fc receptors [126,127]. Together, these observations underscore the importance of future studies to account for the complex interplay among distinct antibody isotypes and Fc receptors in modulating flavivirus immunity and pathogenesis. Determining whether IgA and other non-IgG isotypes mitigate or potentiate antibody-associated disease *in vivo* will inform strategies to improve the safety and efficacy of antibody-based countermeasures [128].

A limitation of our study is that we did not evaluate the *in vivo* protective and pathogenic potential of identified bnAbs, in part due to the lack of an animal model that fully recapitulates dengue immunity and disease [129–131]. Evaluating these properties for IgA antibodies in existing mouse models is especially challenging as they do not express FcαRI homologs [132]. Moreover, rapid IgA clearance in mice [133,134] likely necessitates IgA deglycosylation to improve stability [109], which limits the biological relevance of these animal models for assessing protective or pathogenic functions of IgA antibodies in their native form. Thus, cohort studies similar to those that have defined IgG-associated correlates of protection or disease [12,13,17,21] would be most informative.

Another limitation is that we analyzed antibody repertoires from a relatively small donor sample size. Additionally, because our primary goal was to discover bnAbs, we focused on antibodies encoded by transiently circulating plasmablasts, which often display neutralization breadth and potency. Although there is functional overlap between the DENV-specific plasmablast antibody repertoire with that of memory B cell and long-lived plasma cell subsets [47], future studies will need to determine whether the bnAbs we identified here contribute to durable immunity.

Finally, we acknowledge that by performing single point neutralization assays without normalizing to antibody concentration in the first round of screening, we may have missed some neutralizing antibodies due to low expression. However, we found that neutralizing activity was not simply explained by expression level: 6 out of the 25 highest expressing antibodies ($> 575$ μg/ml) failed to neutralize any virus while 15 out of 25 of the lowest-expressing antibodies ($< 62$ μg/ml) neutralized one or more viruses by $> 50\%$ (S2 Fig). These results suggest that the majority of antibodies screened were tested at concentrations sufficient to identify potently neutralizing antibodies. Moreover, this approach succeeded in achieving our goal of discovering novel flavivirus bnAbs, including the first non-IgG isotype that can potently and broadly neutralize DENV1-4 and ZIKV.

## Methods

### Ethics statement

The study's use of samples from human donors with acute DENV and/or ZIKV infection was approved by the Stanford University Administrative Panel on Human Subjects in Medical Research (Protocol #35460) and the Fundación Valle del Lili Ethics committee in biomedical research (Cali, Colombia). All participants, their parents, or legal guardians provided written informed consent, and subjects 6 years of age and older provided assent.

## Cohort samples

We collected blood samples from individuals who presented with symptoms compatible with dengue between 2016 and 2017 to the Fundación Valle del Lili in Cali, Colombia. Each blood sample was centrifuged to separate serum and peripheral blood mononuclear cells (PBMCs). Sera was stored at -80°C and corresponding PBMCs were cryopreserved and stored in liquid nitrogen. Cohort details have been previously described [41,42].

## Cell lines

ExpiCHO-S Cells (Cat# A29127; ThermoFisher Scientific, Waltham MA) were cultured in ExpiCHO Expression Medium (Cat# A2910001; ThermoFisher Scientific) and maintained at 37°C in 8% CO2 on a platform rotating at 125 rpm with a rotational diameter of 19 cm. They were subcultured according to the manufacturer's instructions. HEK-293T/17 cells (Cat# CRL-11268, ATCC, Manassas, VA) and Vero-C1008 cells (Cat# CRL-1586, ATCC) were maintained in DMEM (Cat# 11965118; ThermoFisher Scientific) supplemented with 7% fetal bovine serum (FBS) (Cat# 26140079, lot 2358194RP, ThermoFisher Scientific) and 100 U/mL penicillin-streptomycin (Cat# 15140–122; ThermoFisher Scientific). Raji cells stably expressing DCSIGNR (Raji-DCSIGNR) [135] (provided by Ted Pierson, NIH), K562 cells (Cat# CCL-243, ATCC), and U937 cells (Cat# CRL-1593.2, ATCC) were maintained in RPMI 1640 supplemented with GlutaMAX (Cat# 72400–047; ThermoFisher Scientific), 7% FBS and 100 U/mL penicillin-streptomycin. C6/36 cells (Cat# CRL-1660, ATCC) were maintained in EMEM (Cat# 30–2003, ATCC) supplemented with 10% FBS at 30°C in 5% CO2. All cell lines were maintained at 37°C in 5% CO2 unless otherwise stated.

## Preparation of cells for single-cell RNA sequencing

Cryopreserved PBMCs were thawed quickly in a 37°C water bath and transferred to a 50 mL conical tube. Thirty mL of RPMI 1640 supplemented with 10% FBS (no antibiotics) was added to the cells dropwise while gently swirling. Cells were counted and CD19+ B cells were isolated using the EasySep Human Pan-B cell enrichment kit (Cat# 19554, StemCell Technologies, Vancouver, Canada) according to the manufacturer's instructions. The resulting cells were incubated in a cocktail containing a live/dead stain (Cat# L34957, Thermo Scientific) and fluorescently labeled antibodies for CD20-eFluor450 (Cat# 48-0209-42, Invitrogen, Waltham, MA), CD38-FITC (Cat# 303504, Biolegend, San Diego, CA), CD27-PE-Cy7 (Cat# 25-0271-82, Invitrogen), CD19-APC (Cat# 555415, BD Biosciences, Franklin Lakes, NJ), CD3-APC-Cy7 (Cat# 300318, Biolegend), CD8-APC-Cy7 (Cat# 344714) and CD14-APC-Cy7 (Cat# 301820) for 30 min at 4°C. Stained cells were washed twice in FACS wash buffer (10% FBS in PBS) and strained through FACS tubes with strainer caps (Cat# 352235, BD Biosciences). The cells were analyzed on a BD FACS Aria flow cytometer to assess the purity of B cells (CD19+) and determine the fraction of cells that were plasmablasts (CD3$^-$, CD8$^-$, CD14$^-$, CD19$^{mid\ to\ hi}$, CD20$^-$, CD27$^+$, CD38$^+$). In pilot experiments, we found that when we sorted samples with fewer than 30,000 plasmablasts, we had difficulty recovering enough cells to achieve the required density and volume for subsequent processing on the 10X Genomics chip. We therefore chose to enrich plasmablasts by FACS only if the sample met two criteria: 1) the fraction of B cells that were plasmablasts was <10%, and 2) the total number of plasmablasts in the sample was > 30,000. Based on these criteria, only one donor's sample (002) was enriched for plasmablasts via FACS; magnetically enriched CD19+ B cells from the remaining samples (001, 012, 014) were processed for scRNAseq without further enrichment.

The cells were prepared for RNA library generation using the Chromium Next GEM Single Cell 5' Library and Gel Bead Kit v1.1 (Cat# PN-1000167, 10X Genomics, Pleasanton, CA)

according to the manufacturer's instructions. A library enriched for variable regions of B cell receptors (BCR library) was generated using the Chromium Single Cell V(D)J Enrichment Kit, Human B Cell (Cat# PN-1000016, 10X Genomics) and the global gene expression library (GEX library) was generated using the Chromium Single Cell 5' Library Construction Kit (Cat# PN-1000020, 10X Genomics), both according to the manufacturer's instructions. Both libraries from the sample from donor 014 (D014) were sequenced on an Illumina HiSeq. The libraries for the samples D001 (donor 001), D002 (donor 002), and D012 (donor 012) were sequenced on an Illumina NovaSeq 6000. Sequencing data were demultiplexed and aligned to the human transcriptome GRCh38-2020-A using cellranger (10X Genomics) version 5.0.1 (D001, D002, D012) or 5.0.0 (D014), which also identified the isotype of each antibody. The "filtered" cellranger output was then passed to *partis* for paired heavy/light chain clustering and annotation with default parameters [44]. This included the default *partis* disambiguation of incomplete and ambiguous heavy/light pairing information, which for instance resolved an atypically large number of droplets in D014 with reads from more than one cell. After grouping all sequences from an individual donor into clonal families, *partis* estimated the V, D, and J gene segments that composed the naive antibody sequence. B cell subtypes were identified using previously described gene markers [49] in the AUCell package (1.12.0). Isotype annotations were taken from the cellranger output.

## Selection of candidate bnAbs from single-cell RNA sequencing data

The variable regions of the paired heavy and light chain sequences were grouped into clusters based on inferred shared ancestry (clonal families) using *partis*. This method first groups together sequences stemming from the same rearrangement event for each chain separately, using a combination of inferred ancestral sequences and likelihood calculation with hierarchical agglomeration. It then refines these clusters using heavy/light chain pairing information. Further details are described in [51]. B cell isotype was determined by aligning sequences to known constant region genes, and selecting the best match. B cell subtype was determined from gene expression data by using the AUCell package [136] to categorize each cell's expression profile by similarity to a set of reference genes that are highly up or downregulated for each subtype (gene sets in **S4 Table**). For the first round of screening intended to find families that encode bnAbs, we selected the largest clonal families from each donor excluding those in which the mean somatic hypermutation (measured by nucleotide sequence) was below 2%. Within the selected families we selected 1–2 sequences that had the lowest Hamming distance to consensus (i.e. the sequence consisting of the most common amino acid present at each position), excluding those that were not encoded by plasmablasts. The selected antibodies were screened for their ability to neutralize DENV1-4 and ZIKV (described below) and those that neutralized >50% of infection of 3 or more viruses were considered "hits". We initiated a second round of screening of antibodies from clonal families that had produced hits in the first round. Within each family we selected antibodies in ascending order of Hamming distance to the consensus, again excluding those that were not encoded by plasmablasts.

## Expression of recombinant antibodies

Heavy and light chain constructs for recombinant MZ4 IgG1 expression have been described previously [33] and were provided by Shelly Krebs (Walter Reed Army Institute of Research). For other antibodies, heavy and light chain variable regions were synthesized (Twist Bioscience, South San Francisco, CA). Variable region sequences for newly identified antibodies were obtained from our scRNAseq data; those for control antibodies were determined based on the protein database (PDB) entries 4UT9 (EDE1-C10), 4UTA (EDE1-C8), 4UT6

(EDE2-B7), 4UTB (EDE2-A11), 7BUD (SIgN-3C), and 3N9G (CR4354) and codon-optimized for gene synthesis. All variable regions were cloned into the expression vectors provided by Patrick Wilson (University of Chicago): AbVec-hIgG1 (GenBank accession # FJ475055), AbVec-hIgKappa (GenBank accession# FJ475056) and AbVec-hIgLambda (GenBank accession # FJ517647), respectively. The variable regions were synthesized with adaptor sequences overlapping their respective vectors. The adaptor sequence that was appended to the 5' end was the same for all vectors: TAGTAGGAACTGCAACCGGTT. The sequence appended to 3' ends was specific to each vector: for AbVec heavy: CGGTCGACCAAGGGCCCATCGG, for AbVec kappa: CGTACGGTGGCTGCACCATC, and for AbVec lambda: GGTCAGCCCAA GGCCAACCCCACTGTCACTCTGTTCCCACCCTCGAGTGAGGAGCTTCAAGC. Heavy, kappa, and lambda vectors were linearized by digestion with SalI/AgeI, BsiWI/AgeI, and XhoI/AgeI, respectively as described [137]. Synthesized fragments and linearized vectors were ligated using NEBuilder HiFi DNA Assembly Master Mix (Cat# E2612L, New England Bio-labs, Ipswich, MA) according to the manufacturer's instructions.

IgA1 heavy chains were generated by cloning the variable regions of selected antibodies into the expression vector pFUSEss-CHIg-hA1 (Cat# pfusess-hcha1, Invivogen, San Diego, CA). Variable regions of the antibody coding sequences were PCR amplified using the IgG1 heavy chain expression plasmid as a template and custom primers that appended an EcoRI site and an NheI site at the 5' and 3' ends respectively. Primer sequences were as follows: for F25.S02 GTACACGAATTCGCAGGTGCAGCTGGTGC (forward) and GACTCTGCTAGC TGAGGAGACGGTGACC (reverse); for EDE1-C10 GTACACGAATTCGGAGGTCCAACT TGTTG (forward) and GACTCTGCTAGCAGAGCTTACGGTTACG (reverse); and for SIgN-3C GTACACGAATTCGGAAGTACAACTGGTGC (forward) and GACTCTGCT AGCTGAACTAACAGTTACCAG (reverse). The PCR amplicons and the vector were digested with EcoRI and NheI and the resulting fragments were ligated using T7 DNA ligase (Cat# M0318, New England Biolabs).

All AbVec antibody expression plasmids (IgG1-heavy, kappa, and lambda) were confirmed by Sanger sequencing (Fred Hutch Genomics Core) using the primer "AbVec sense": GCTTCG TTAGAACGCGGCTAC. IgA1 expression plasmids were confirmed by whole plasmid nanopore sequencing (Plasmidsaurus, Eugene, OR). To produce IgG1 and monomeric IgA1, heavy and light chain expression vectors were co-transfected into cultures of ExpiCHO-S cells at 0.8 ng/mL total DNA concentration at 1:1 mass ratio using OptiPro serum free medium (Cat#12309, Gibco) and Expifectamine CHO Transfection Kit (Cat# A29130, Gibco) according to the manufacturer's instructions. To produce IgA1 dimers, plasmids encoding heavy, light, and joining chain (Cat# pUNO4-hJCHAIN, InvivoGen) were co-transfected at 0.8 ng/mL total DNA concentration at 1:1:1 mass ratio using the same medium and transfection reagents. Supernatant containing secreted antibodies was collected 8 days post transfection, centrifuged at 3220 x g for 10 minutes and filtered through a 0.45 µm Steriflip filter (Cat# SE1M003M00, Millipore-Sigma).

## Purification of antibodies

Recombinant IgG1 produced in transfected ExpiCHO-S cells was purified using MabSelect Sure LX protein A agarose beads (Cat# 17-5474-01, Cytiva Life Sciences, Marlborough, MA) according to the manufacturer's instructions. Recombinant IgA1 produced in ExpiCHO-S cells as described above was purified using protein M agarose beads (Cat# gel-pdm-2, Invivo-Gen US) according to the manufacturer's instructions. IgA1 multimers were separated from monomers via size exclusion chromatography on a HiLoad 16/600 Superdex 200 pg column using 70 mL PBS as the eluate. A monomeric IgA1 antibody (Cat# 31148, ThermoFisher) was used as a standard for SDS-PAGE and as a negative control for ADE assays as indicated.

The hybridoma D1-4G2-4-15, which expresses the antibody 4G2, was obtained from ATCC (Cat# HB-112). The hybridoma was expanded and IgG was purified from culture supernatant by the Fred Hutchinson Cancer Center Antibody Technology Core. The purified antibody was conjugated to APC using the Lighting-Link APC-conjugation kit (Cat# ab201807, Abcam) according to the manufacturer's instructions and used to detect intracellular E protein in assays using fully infectious DENV. Unconjugated, purified 4G2 antibody was used in ELISA experiments.

## ELISA to quantify antibodies

96 well absorbent plates (Cat# 3361, Corning Inc.) were coated overnight with 50 μL/well of 25 mg/mL goat antibody raised against human IgG, IgA, and IgM (Cat# I1761 Sigma-Aldrich). The next day wells were washed with 200 μL wash buffer (0.1% Tween-20 in PBS) and blocked with 200 μL of 3% nonfat milk in PBS for 1 hour. Wells were washed once in wash buffer and 50 μL of sample was added to each well. Samples were incubated for 2 hours at room temperature on a rocker. After incubation, wells were washed 3 times in wash buffer, followed by addition of 50 μL of peroxidase-conjugated goat anti-human IgG secondary antibody (Cat# A0170, ThermoFisher) diluted 1:50,000. The secondary antibody was incubated for 1 hour at room temperature on a rocker. Wells were washed 3 times in wash buffer, received 50 μL of TMB (Cat# 34028, ThermoFisher), and were incubated at room temperature until a color change was apparent. The reaction was stopped with 50 μL of 1N HCl and absorbance at 450 nm was read on SpectraMax i3x plate reader (Molecular Devices). The IgG1 concentrations of unknown samples were measured by comparison to wells containing known concentrations of purified CR4354 IgG1.

## Production of single-round infectious reporter virus particles

Reporter virus particles of DENV1, DENV2, ZIKV, and WNV were produced by co-transfection of HEK-293T/17 cells with (i) a plasmid expressing a WNV subgenomic replicon encoding GFP in place of structural genes [138], and (ii) a plasmid encoding C-prM-E structural genes from the following viruses: DENV1 Western Pacific-74 (WP-74) [139], DENV1 16007 [140], DENV2 16681 [139], DKE-121 [71], WNV NY99 [138], and ZIKV H/PF/2013 [141]. Briefly, 8 x 10^5 HEK-293T/17 cells were plated in each well of a 6-well plate, The following day each well was co-transfected with 1 μg of replicon-encoding plasmid and 3 μg of C-prM-E-encoding plasmid using Lipofectamine 3000 (Cat# L3000-015; ThermoFisher Scientific) according to the manufacturer's instructions. Four hours post-transfection, media was replaced with low-glucose DMEM (Cat# 12320–032; ThermoFisher Scientific) containing 7% FBS and 100 U/mL penicillin-streptomycin (i.e. low-glucose DMEM complete) and cells were transferred to 30°C in 5% CO2. Virus-containing supernatant was harvested twice per day at days 3 through 8 post-transfection and centrifuged at 700 x g for 5 min. The clarified supernatant was passed through a 0.45 μm Steriflip filter (Cat# SE1M003M00, Millipore-Sigma, St. Louis, MO), pooled, aliquoted, and stored at -80°C. Reporter virus particles with increased efficiency of prM cleavage were produced as above by co-transfecting plasmids encoding the replicon, structural genes, and human furin (provided by Ted Pierson, NIH) at a 1:3:1 mass ratio.

Reporter virus particles of DENV3 strain CH53489 (Cat# RVP-301; Integral Molecular, Philadelphia, PA) and DENV4 strain TVP376 (Cat# RVP-401; Integral Molecular) were obtained commercially and were produced by co-transfection of the DENV3 or DENV4 CprME plasmid with the DENV2 strain 16681 replicon as previously described [142].

For the RVP binding ELISA described below ELISA (Fig 4A and 4B) and ADE experiments in U937 cells (Fig 7), DENV2 16681 reporter virus particles were concentrated 100X by

ultracentrifugation through 20% sucrose at 166,880 x g for 4 hr at 4°C, resuspended in 1/100 volume of HNE buffer (5 mM HEPES, 150 mM NaCl, 0.1 mM EDTA, pH 7.4), and stored at -80°C.

Infectious titers of reporter viruses were determined by infection of Raji-DCSIGNR cells using 2-fold serial dilutions of virus stocks. At 2 days post-infection, cells were fixed in 2% paraformaldehyde (Cat# 15714S; Electron Microscopy Sciences, Hatfield, PA), and %GFP positive cells quantified by flow cytometry (Intellicyt iQue Screener PLUS, Sartorius AG, Gottingen, Germany).

### Generation of E protein variants

Construction of DENV2 16681 reporter virus variants in which E protein sites were substituted with corresponding ZIKV H/PF/2013 amino acid residues individually or in combination have been previously described [39]. Here, we used similar methods to generate individual alanine mutations. Specifically, the DENV2 16681 CprME expression construct [139] was used as a template for Q5 site-directed mutagenesis (Cat# E0554S; New England Biolabs, Ipswich, MA) and primers generated by NEBaseChanger (New England Biolabs, Ipswich, MA). The entire plasmid was sequenced (Plasmidsaurus, Eugene, OR) to confirm the presence of the desired mutation(s) only.

### E protein and reporter virus particle binding ELISA

Nunc 384-Well Clear Polystyrene Plates (Cat# 164688 ThermoFisher) were coated with 20 μL/well of recombinant E monomers (Cat#DENV2-ENV, Native Antigen Co, Kidlington, United Kingdom) at 3 μg/mL or 20 μL/well of antibody 4G2 at 50 μg/mL overnight. The next day plates were washed once with 50 μL wash buffer (0.05% Tween-20 in PBS) and blocked with 50 μL of blocking buffer (3% nonfat milk, Cat# 20–241 Apex Bioresearch products, in PBS) at 37°C for 45 min. Blocking buffer was aspirated from wells that had received 4G2 and replaced with 20 μL of 100X concentrated reporter virus particles diluted 1:1 in blocking buffer. Wells that had received E monomers were left in blocking buffer and plates were incubated at 37°C for 45 min. Wells were washed 3 times with 50 μL of wash buffer, received 30 μL of primary antibody at 100 μg/mL, and were incubated at 37°C for 45 min. Wells were washed 6 times with 50 μL wash buffer, received 30 μL of mouse anti-human antibody (Cat# 05–4220, ThermoFisher) at 1 μg/mL, and were incubated at 37°C for 45 min. Finally, wells were washed 6 times with 50 μL wash buffer, received 30 μL of TMB (Cat# 34028 ThermoFisher), and were incubated at room temperature until a color change was apparent. The reaction was stopped with 15 μL of 1N HCl and absorbance at 450 nm was read on SpectraMax i3x plate reader (Molecular Devices, San Jose, CA)

### Binding screen against alanine library

We screened binding of antibodies F25.S02 and F05.S03 to a DENV2 16681 library where each prM/E polyprotein residue was mutated to alanine (or alanine residues to serine) [81]. In total, 559 sequence confirmed DENV2 mutants (99.6% coverage of the prM/E protein) were arrayed into 384-well plates (one mutation per well). The optimal screening condition was determined using an independent immunofluorescence titration curve against wild-type prM/E expressed in HEK293T cells to ensure that signals were within the linear range of detection and that signal exceeded background by at least 5-fold. F25.S02 and F05.S03 bound sufficiently well for screening only when the prM/E expression plasmid was co-transfected with a furin expression plasmid to enhance cleavage of prM. Thus, for antibody screening, plasmids encoding the DENV protein variants were individually co-transfected with furin expression plasmid into

HEK-293T cells and expressed for 22 hr before incubation with purified IgG1 antibodies (0.1–2.0 μg/mL) diluted in 10% normal goat serum (NGS) (Sigma-Aldrich, St. Louis, MO) in PBS plus calcium and magnesium (PBS++).

Antibodies were detected using 3.75 μg/mL Alexa Fluor 488-conjugated secondary antibody (Jackson ImmunoResearch Laboratories) in 10% NGS. Cells were washed three times with PBS++ followed by 2 washes in PBS, then fixed in 4% paraformaldehyde (Electron Microscopy Sciences), washed in PBS, and resuspended in Cellstripper (Cat# 25-056-CI, Corning Inc, Corning, NY) plus 0.1% BSA (Sigma-Aldrich). Mean cellular fluorescence was detected by flow cytometry (Intellicyt iQue Screener PLUS, Sartorius AG).

Antibody reactivity against each mutant was calculated relative to reactivity with wild-type prM/E, by subtracting the signal from mock-transfected controls and normalizing to the signal from wild-type protein-transfected controls. The entire library data for each antibody was compared to control antibodies. Mutations were identified as critical to the antibody epitope if they did not support reactivity of the test antibody, but supported reactivity of other control antibodies. This counter-screen strategy facilitates the exclusion of DENV prM/E protein mutants that impact folding or expression.

## Neutralization and antibody-dependent enhancement assays using single-round infectious reporter virus particles

All neutralization and ADE assays using the following strains were performed with reporter virus particles: DENV1 West-Pac 74, DENV1 16007, DENV2 16681, DENV3 CH53489, DENV4 TVP376, DKE-121, ZIKV H/PF/2013, WNV NY99. For experiments with DENV reporter viruses, except for Fig 3, which tested the entire panel of strains listed above in addition to fully infectious viruses described in the next section, neutralization assays were performed using a condensed panel of commonly used strains of reporter viruses representing each serotype (DENV1 West-Pac 74, DENV2 16681, DENV3 CH54389, DENV4 TVP376). Depending on the assay, stocks of reporter virus particles diluted to 5–10% final infectivity were incubated with either heat-inactivated serum (56˚C for 30 min), 1/10 diluted ExpiCHO-S cell supernatant containing recombinant IgG1, or 5-fold serial dilutions of purified monoclonal antibodies for 1 hr at room temperature before addition of 2e5 Raji-DCSIGNR cells (neutralization assays), K562 cells (ADE assays), or U937 cells (ADE assays). After incubation for 2 days at 37˚C, cells were fixed in 2% paraformaldehyde and GFP positive cells were quantified by flow cytometry (Intellicyt iQue Screener Plus, Sartorius AG). For experiments using single dilutions of serum or ExpiCHO-S cell supernatant, infection was normalized to conditions without serum/supernatant and expressed as % infection of the untreated condition. For experiments using serial dilutions of serum or of purified monoclonal antibodies, infection was normalized to conditions without serum/antibody and analyzed by non-linear regression with a variable slope and the bottom and top of the curves constrained to 0% and 100%, respectively (Graph-PadPrism v8, GraphPad Software Inc). Results from experiments using serially diluted serum were reported as the reciprocal dilution at which 50% of infection was neutralized (NT50). Results from experiments using serially diluted purified antibodies were reported as the concentration at which 50% of infection was neutralized (IC50).

## Production and neutralization of fully infectious virus

DENV1 UIS 998 (isolated in 2007, Cat# NR-49713), DENV2 US/BID-V594/2006 (isolated in 2006, Cat# NR-43280), DENV3/US/BID- V1043/2006 (isolated in 2006, Cat# NR-43282), DENV4 strain UIS497 (isolated in 2004, Cat# NR-49724) were obtained from BEI Resources (Manassas, VA). Viral stocks were expanded by infecting 70% confluent C6/36 cells and virus-

containing supernatant was collected and pooled at days 3 to 8 post infection. DENV4 H241 (isolated 1956, Cat# TVP17463) was obtained from the World Reference Center for Emerging Viruses and Arboviruses (WRCEVA) at the University of Texas Medical Branch (Galveston, TX). The seed stock was expanded by infecting 90% confluent Vero cells and virus-containing supernatant was collected 7 days post infection. All virus-containing supernatants were centrifuged at 500 x g for 5 min, filtered through a 0.45 µm Steriflip filter (Cat# SE1M003M00, Millipore-Sigma), and stored at -80˚C. Viral stocks were titered by infecting 2e5 Raji-DCSIGNR cells with 2-fold serial dilutions. Two days post infection cells were fixed and permeabilized using BD cytofix/cytoperm (Cat# 554717, BD Biosciences) according to the manufacturer's instructions before incubation with APC-conjugated 4G2 (an antibody specific for E protein) for 30 minutes at 4˚C. Cells were washed twice in cytoperm/wash buffer and APC+ positive cells were quantified by flow cytometry.

For dose response neutralization assays using fully infectious virus, stocks were diluted to achieve 5–10% infection in Raji-DCSIGNR cells were incubated with 5-fold serial dilutions of antibodies for 1 hour, then combined with 2e5 Raji-DCSIGNR cells and incubated at 37˚C 5% $CO_2$, before being stained for E protein as described above. IC50 values were calculated as described above for neutralization assays using reporter virus particles.

### Determining Fc receptor expression

K562 cells and U937 cells were washed in FACS wash (FW, 2% FBS in PBS) and resuspended in 50 µL of staining or isotype control antibody and incubated at 4˚C for 30 min. For FcγRII we stained with anti-CD32-FITC (Cat# 60012.FI, StemCell) and corresponding mouse IgG2b-FITC isotype control (Cat# 11-4732-81, ThermoFisher Scientific). For FcαRI we stained with anti-CD89/-PE (cat# 555686, BD Biosciences) and corresponding mouse IgG1-PE isotype control (cat# 12-4714-42, ThermoFisher Scientific). Cells were washed twice in FW and analyzed by flow cytometry.

### Supporting information

**S1 Fig. Serum neutralizing activity against flaviviruses.** Serum samples from 38 cohort participants with the indicated age and DENV and/or ZIKV acute exposures collected at the time point(s) shown were diluted either 1:240 (expt1) or 1:300 (expt2) and tested for their ability to neutralize the indicated reporter viruses in two independent experiments. Bottom rows indicate control antibodies, which include human convalescent sera to DENV (BEI Resources NR-50232) or ZIKV (BEI Resources NR-50752) and monoclonal antibodies (mAb) E60 [143], ZV-67 [144], CR4354 [54], and EDE1-C10 [28]. The percent neutralizing activity shown under each virus column is normalized to infection in the absence of antibody. Heatmap colors represent neutralizing activity of at least 50% as indicated in the key under the table. We selected corresponding PBMC samples from the donors and time points highlighted in blue under the 'Days post-fever' column for single-cell RNA sequencing to isolate monoclonal antibodies.
(TIF)

**S2 Fig. Neutralization profiles of IgG1 transfection supernatants from rounds 1 and 2 of screening.** Transfection supernatant containing (**A**) control antibodies, EDE1-C10 [28,31] and CR4354 [54] or antibodies from donors (**B**) 001, (**C**) 012, (**D**) 002, and (**E-G**) 014 indicated in each row was tested for neutralization against DENV1 WP-74, DENV2 16681, DENV3 CH54389, DENV4 TVP376, ZIKV H/PF/2013, and WNV NY99 reporter viruses. The second column displays the concentration of IgG1 detected in each crude supernatant as

determined by ELISA, displayed as a blue heatmap according to the key. The supernatant composed 1/10 of the volume of each neutralization assay, so the final concentration of antibody present in each assay was 1/10 the value displayed. The red heatmap displays the percent neutralization of each virus normalized to infection in the absence of antibody, as indicated in the key (only values >25% are highlighted in each panel). Antibodies were named based on the source of the antibody in the format DXX.FYY.SZZ, where XX is the donor number, YY is the clonal family within the donor ranked by decreasing size, and ZZ is assigned by the chronological order in which antibodies from the family were produced. Antibodies whose names are left aligned were screened in round 1, which was intended to screen many different families. Antibodies that were considered hits due to the breadth and/or potency of their neutralization in round 1 are shown in bold font. For round 2 we selected additional antibodies, shown indented and italicized, from the clonal families of hits identified in round 1.
(TIF)

**S3 Fig. Effect of virion maturation state on bnAb activity. (A)** The indicated antibodies were tested against DENV2 16681 (blue) or ZIKV H/PF/2013 (black) reporter virus particles prepared either under standard conditions (solid circles and lines) or in the presence of excess furin (open circles and dashed lines). Data were obtained from two independent experiments, each performed in duplicate wells. Data points and error bars represent the mean infection and standard deviation of the four total replicates, respectively. **(B)** The table displays the mean IC50 values at which the indicated antibodies neutralized the indicated forms of DENV2 and ZIKV in dose response neutralization curves as shown in (A).
(TIF)

**S4 Fig. Effect of antibody valency on neutralizing activity.** We tested monovalent Fab (open circles and dashed lines) or bivalent IgG1 (solid circles and lines) versions of antibodies **(A)** F25.S02, **(B)** F09.S05, **(C)** F05.S03, **(D)** EDE1-C10, and **(E)** SIgN-3C against DENV2 16681 (black) or ZIKV H/PF/2013 (blue) reporter virus particles. Dose-response neutralization curves shown are from two independent experiments, each performed in duplicate wells. Data points and error bars represent the mean infection and standard deviation of the four total replicates, respectively.
(TIF)

**S5 Fig. Purity of IgA1 antibody preparations.** Graphs on the left display absorbance profiles (at 280 nm) of eluates from size-exclusion chromatography (SEC), which was used to separate monomeric and polymeric IgA1. Images on the right display SDS-PAGE gels to assess purity of preparations. Eluates from SEC were collected in 2 mL fractions and the fractions indicated were collected, pooled, and concentrated to obtain purified monomers and polymers. SDS-PAGE was run on non-reduced (left half) and reduced (right half) samples of each type of antibody. Each half of a gel has one well containing a commercially purchased IgA1 isotype control (IgA Std). Each half also has wells containing two types of IgA1 monomers. The first was produced as monomers, i.e in the absence of a J chain expression plasmid (Mono). The second were produced in a transfection that included a J chain expression plasmid and they were separated from polymers via SEC (S-Mono). Both types of monomers appeared similar by SDS-PAGE, but for simplicity all experiments were performed using the monomers produced in the absence of J chain.
(TIF)

**S6 Fig. ADE profile of bnAbs in K562 cells. (A)** Serial dilutions of IgG1 antibodies indicated in the key were complexed with DENV1 WP-74, DENV2 16681, DENV3 CH53489, DENV4 TVP376, and ZIKV H/PF/2013 reporter virus particles prior to infection of K562 cells, which

express FcγRIIa but not FcαRI. Dose-response ADE profiles of antibodies that do or do not neutralize ZIKV in addition to DENV1-4 are shown in top and bottom panels, respectively. Data points and error bars indicate the mean and range of infection in duplicate wells, respectively. Graphs shown are representative of 4–5 independent experiments. **(B)** IgG1 and monomeric IgA1 forms of F25.S02 (top row), EDE1-C10 (middle row) or SIgN-3C (bottom row) were tested either individually or mixed at the indicated ratios by mass before serial dilution and pre-incubation with DENV1 WP-74 (left) or DENV4 TVP376 (right) reporter virus particles. Virus-antibody complexes were then used to infect K562 cells. Addition of isotype control IgA1 to IgG1 forms of each bnAb was included as controls. The experiment was performed in three biological replicates, each in duplicate wells. The data points and the error bars represent the means and the range of the duplicate wells, respectively, of one representative experiment. (TIF)

**S7 Fig. Fc receptor expression profile of K562 and U937 cells.** Histograms display the fluorescence intensity of K562 (top row) or U937 (bottom row) cells stained for the indicated Fc receptors. Histograms are normalized to the modal cell count. The isotype control was conjugated to the same fluorophore and used at the same concentration as anti-FcγRIIa or anti-FcαRI antibody on the same population of cells. (TIF)

**S1 Table. Heatmaps of IC50 values against DENV1-4 and ZIKV reporter virus particles for previously published bnAbs, novel category 1 bnAbs (neutralize DENV1-4 and ZIKV), and novel category 2 bnAbs (neutralize DENV1-4 but not ZIKV).** For each virus, the value reported is the arithmetic mean IC50 from at least three independent experiments performed in duplicate. *Geometric mean IC50 for all neutralized viruses, i.e. values >10,000 ng/ml (the highest antibody concentration tested) were omitted. All antibodies were isolated from donor 014 except for F15.S01, which was isolated from donor 012. (TIF)

**S2 Table. Genetic characteristics of broadly neutralizing antibodies whose IC50 values are displayed in S1 Table.** Bold = chosen for detailed characterization; blue = non-IgG isotype; ? = insufficient sequence coverage of constant gene to determine the antibody's isotype; pb = plasmablast. (TIF)

**S3 Table. Antibody binding reactivity to DENV2 16681 E protein alanine scanning mutagenesis library.** Mean percentage and range of binding reactivity to alanine mutant relative to wild type DENV2 from at least two independent experiments. (TIF)

**S4 Table. Expression of reference genes used to determine B cell subset.** (TIF)

**S1 Data. Excel spreadsheet containing, in separate sheets, the underlying numerical data used to generate Figs 1A, 2A, 2B, 3A-3D, 4A, 4B, 5A–5E, 6A, 6B, 7A–7C, S1–S4, S6A and S6B and S1 Table.** (XLSX)

## Acknowledgments

We thank cohort participants and staff; John McNevin, Andrew Berger, and Brian Raden, for assistance with cell sorting; Adam Waickman for advice on scRNAseq analysis; Patrick Wilson

for providing IgG1 expression vectors, Ted Pierson for providing Raji-DCSIGNR cells and constructs for reporter virus production; Leah Homad and Andrew McGuire for assistance with polymeric IgA1 purification; Michael Diamond for providing E60 and ZV-67 antibodies; Shelly Krebs for providing expression constructs for antibody MZ4; and Dennis Burton for providing antibody PGT121.

Fully infectious DENV1-4 isolates were obtained through BEI Resources, NIAID, NIH, as part of the World Reference Center for Emerging Viruses and Arboviruses program (WRCEVA).

Molecular graphics and analyses of DENV2 E dimer were performed with UCSF ChimeraX, developed by the Resource for Biocomputing, Visualization, and Informatics at the University of California, San Francisco, with support from National Institutes of Health R01-GM129325 and the Office of Cyber Infrastructure and Computational Biology, National Institute of Allergy and Infectious Diseases.

## Author Contributions

**Conceptualization:** Leslie Goo.

**Formal analysis:** Jay Lubow, Lisa M. Levoir, Duncan K. Ralph, Laura Belmont, Maya Contreras, Catiana H. Cartwright-Acar, Caroline Kikawa, Shruthi Kannan, Edgar Davidson, Leslie Goo.

**Funding acquisition:** Shirit Einav, Frederick A. Matsen IV, Leslie Goo.

**Investigation:** Jay Lubow, Lisa M. Levoir, Duncan K. Ralph, Laura Belmont, Maya Contreras, Catiana H. Cartwright-Acar, Caroline Kikawa, Shruthi Kannan, Edgar Davidson, Leslie Goo.

**Methodology:** Jay Lubow, Lisa M. Levoir, Duncan K. Ralph, Laura Belmont, Shruthi Kannan, Edgar Davidson, Benjamin J. Doranz, Leslie Goo.

**Project administration:** Veronica Duran, David E. Rebellon-Sanchez, Ana M. Sanz, Fernando Rosso.

**Resources:** David E. Rebellon-Sanchez, Ana M. Sanz, Fernando Rosso, Shirit Einav.

**Software:** Duncan K. Ralph.

**Supervision:** Fernando Rosso, Benjamin J. Doranz, Shirit Einav, Frederick A. Matsen IV, Leslie Goo.

**Validation:** Jay Lubow, Lisa M. Levoir, Laura Belmont, Maya Contreras, Catiana H. Cartwright-Acar, Caroline Kikawa, Leslie Goo.

**Visualization:** Jay Lubow, Duncan K. Ralph, Laura Belmont, Maya Contreras, Catiana H. Cartwright-Acar, Caroline Kikawa, Leslie Goo.

**Writing – original draft:** Jay Lubow, Leslie Goo.

**Writing – review & editing:** Jay Lubow, Duncan K. Ralph, Laura Belmont, Edgar Davidson, Shirit Einav, Frederick A. Matsen IV, Leslie Goo.

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
