## [Decision Letter · Decision Letter 0]

7 Jul 2023

Dear Dr Goo,

Thank you very much for submitting your manuscript "Single B cell transcriptomics identifies multiple isotypes of broadly neutralizing antibodies against flaviviruses" for consideration at PLOS Pathogens. As with all papers reviewed by the journal, your manuscript was reviewed by members of the editorial board and by several independent reviewers. In light of the reviews (below this email), we would like to invite the resubmission of a significantly-revised version that takes into account the reviewers' comments.

I am returning your manuscript with two reviews. The reviewers agreed that the identification and evaluation of a flavivirus specific IgA for protection and possible ADE is an under-studied area of research in the field. However, the reviewers outlined major issues as you will see below. Since there are recommendations for significant text revisions and additional experiments, I recommend Major Revisions. We are looking forward to receiving your revised manuscript.

Please pay particular attention to the reviewer suggestions outlined below and address them appropriately.

• The text needs to be focused and clarified, particularly related to the B cell sorting and bioinformatics analysis.

• Please address whether polymeric (dimeric) IgA can enhance infection and evaluate IgA and IgG competition in another cell line.

We cannot make any decision about publication until we have seen the revised manuscript and your response to the reviewers' comments. Your revised manuscript is also likely to be sent to reviewers for further evaluation.

Sincerely,

Julie Fox, Ph.D.

Guest Editor

PLOS Pathogens

Sonja Best

Section Editor

PLOS Pathogens

Kasturi Haldar

Editor-in-Chief

PLOS Pathogens

orcid.org/0000-0001-5065-158X

Michael Malim

Editor-in-Chief

PLOS Pathogens

orcid.org/0000-0002-7699-2064

I am returning your manuscript with two reviews. The reviewers agreed that the identification and evaluation of a flavivirus specific IgA for protection and possible ADE is an under-studied area of research in the field. However, the reviewers outlined major issues as you will see below. Since there are recommendations for significant text revisions and additional experiments, I recommend Major Revisions. We are looking forward to receiving your revised manuscript.

Please pay particular attention to the reviewer suggestions outlined below and address them appropriately.

• The text needs to be focused and clarified, particularly related to the B cell sorting and bioinformatics analysis.

• Please address whether polymeric (dimeric) IgA can enhance infection and evaluate IgA and IgG competition in another cell line.

Reviewer's Responses to Questions

**Part I - Summary**

Reviewer #1: This manuscript by Lubow and colleagues report the identification of broadly neutralising antibodies against flaviviruses, in particular dengue viruses (DENVs) and that the IgA1 version of this antibody mediated neutralisation without enhancement, despite dilution. The authors conducted extensive studies to identify such antibodies and mapped their binding sites before exploring how IgG and IgA versions of antibodies with identical Fabs impact both DENV neutralisation and infection enhancement. This work adds to the very limited body of information on IgA antibodies against DENVs and suggest the potential of this type of antibody for therapeutic application.

Reviewer #2: This manuscript by Lubow et al. use scRNA-seq to identify broadly neutralizing antibodies (bnAbs) against dengue virus (DENV1-4) and Zika virus (ZIKV). The research utilizes an unbiased single-cell RNA sequencing (scRNAseq) approach of B cells, which allowed them to identify new bnAbs. This approach allowed the researchers to identify an IgA1 antibody (F25.S02) that exhibits potent cross-neutralizing activity against DENV1-4 and ZIKV. F25.S02 not only maintains its potency when expressed as IgG1 but also has unique binding and neutralization determinants compared to previously known classes of bnAbs. The study further shows that the IgA1 versions of bnAbs can inhibit IgG1-mediated antibody-dependent enhancement (ADE) in a dose-dependent manner. IgG antibodies have been the focus of flavivirus immunity studies, and these findings suggest an underappreciated role of flavivirus-specific IgA antibodies in infection and immunity. While the finding of the competitive role of IgA1 in preventing IgG induced ADE is interesting, there are some major gaps in how IgA is functioning, and the current data is a bit preliminary. Moreover, the B cell analyses in Figure 2 are very generalized and lack rigor in analysis. Lastly, the manuscript poorly references the data within the figures, making it difficult to connect data to the text.

**Part II – Major Issues: Key Experiments Required for Acceptance**

Reviewer #1: The manuscript can greatly benefit from a revised narrative that focuses on the knowledge this study brings to flaviviral or even dengue immunity. In its present form, what knowledge Figures 2 and 3, along with the associated supplementary figures bring, is unclear. For instance, why was it that only donor 014 displayed the broadest neutralisation profile (lines 190-191)? Without knowing where the authors were going with the narrative, the very detailed description became a distraction. If there is indeed biological insights that can be drawn from this section of the manuscript, the description should be clearly framed to provide both context and interpretation. If not, the details could be best described in the methods or even supplementary section for improved flow of the narrative.

Line115. Not all the viruses used in this neutralisation screen are prototypic. Please revise.

Line 183. Were positive and negative controls included in this screen at a single dilution?

Figure 6B. The difference in IC50 between polymeric and monomeric IgA1 was greatest for F25.S02 against DENV2 and SIgN-3C against DENV3 and ZIKV. In contrast, although the rest also show lower IC50 with polymeric IgA1, they do not appear to be statistically significant. Are there any possible reasons for these findings?

Reviewer #2: 1. It is not clear why the authors sorted out only CD19+ cells in order to find rare populations. The authors make a nice argument for why plasmablasts are a useful population to look at.

2. Line 155, the authors conclude that Donors 002 and 014 have more plasmablasts than donors 001 and 012. While this appears to be true, there are lot of plasmablast clonal expansions in 001 and 012, but are presented on a different scale. It should also be noted that the authors crudely sorted on CD19+ cells and as a result, it’s not surprising there are a lot of naïve B cells.

3. Figure 2 very crudely parses out the distinct B cell subsets, with limited details about the bioinformatics on how this done, fidelity of subsetting, etc. Further interrogation and analyses presented would help with this conclusion.

4. Paper generally poorly references data. For example, in the paragraph starting at line 181, a reference to data is only made once. Paragraph starting at Line 212 doesn’t even link to data in the paper.

5. Fig. S2. The use of crude supernatants for neutralization experiments without standardizing for antibody is not recommended. What if a clone expresses better and as a result, this one is selected for downstream analysis when there is a better antibody in the dataset?

6. Line 200, hard to make this conclusion without knowing the number of mAbs made.

7. Figure 7 and associated text, is IgA+virus able to engage FcaRI? FcaRI is a low affinity receptor, as a result requires cross-linking to induce engagement. Does dimeric (or polymeric) IgA lead to ADE?

8. Does the competition between IgG and IgA occur in the K562 cell line?

**Part III – Minor Issues: Editorial and Data Presentation Modifications**

Reviewer #1: Line 269. There is no fifth DENV serotype. Despite the claims of one of the authors, the papers that examined this DENV strain concluded that it was a divergent DENV4, which is not surprising since this is a sylvatic virus. Please revise.

Lines 310-311 and Fig S1. The viral strains described in the methods section are inconsistent with those reported in Fig S1.

Reviewer #2: 1. Lines 120-121 – the authors mention that” the high-prevalence of cross-serotype neutralizing antibodies likely reflects repeated DENV exposures, as confirmed by IgG avidity testing” and then reference a few papers. The way this is written makes it sound like they are doing this in the study, but its actually data from other studies. I would recommend rewording this to make sure it is clear that this already published data.

2. Line 459, lack of ADE by those mAbs are due to the antibodies being unable to neutralize or to bind altogether? This is a bit confusing given that in the introduction that it was proposed non-neutralizing antibodies are responsible for ADE.

3. Patient information just references two other papers. This should be included within the manuscript.

PLOS authors have the option to publish the peer review history of their article (what does this mean?). If published, this will include your full peer review and any attached files.

Reviewer #1: **Yes: **Eng Eong Ooi

Reviewer #2: No
---

## [Decision Letter · Decision Letter 1]

21 Sep 2023

Dear Dr Goo,

Thank you very much for submitting your manuscript "Single B cell transcriptomics identifies multiple isotypes of broadly neutralizing antibodies against flaviviruses" for consideration at PLOS Pathogens. As with all papers reviewed by the journal, your manuscript was reviewed by members of the editorial board and by several independent reviewers. The reviewers appreciated the attention to an important topic. Based on the reviews, we are likely to accept this manuscript for publication, providing that you modify the manuscript according to the review recommendations.

I am returning your manuscript with two reviews. While the majority of the reviewers concerns have been addressed, there is still one major issue regarding Figure S2 and the lack of antibody quantification.

This applies particularly to the antibody supernatants that failed to potently neutralize any of the viruses tested. Since this is related to a supplementary figure, one option would be to remove the antibodies from the table that failed to neutralize any virus due to the fact that the presence of antibody in the supernatant has not been verified. If the authors wish to keep this data in the manuscript, the antibodies must be purified or quantification by ELISA to support the stated conclusions. Additionally, for antibodies that neutralized one but not all of the viruses tested, it can be assumed that the mAb was expressed. However, the conclusions in the results section should outline the positive neutralization results rather than determining the virus specificity of the mAbs since lack of virus neutralization may be related to insufficient quantity of antibody present in the supernatant.

We are looking forward to receiving your revised submission.

Sincerely,

Julie Fox, Ph.D.

Academic Editor

PLOS Pathogens

Sonja Best

Section Editor

PLOS Pathogens

Kasturi Haldar

Editor-in-Chief

PLOS Pathogens

orcid.org/0000-0001-5065-158X

Michael Malim

Editor-in-Chief

PLOS Pathogens

orcid.org/0000-0002-7699-2064

I am returning your manuscript with two reviews. While the majority of the reviewers concerns have been addressed, there is still one major issue regarding Figure S2 and the lack of antibody quantification.

In my opinion, this particularly applies to the antibody supernatants that failed to potently neutralize any of the viruses tested. Since this is related to a supplementary figure, one option would be to remove the antibodies from the table that failed to neutralize any virus due to the fact that the presence of antibody in the supernatant has not been verified. If the authors wish to keep this data in the manuscript, the antibodies must be purified or quantification by ELISA to support the stated conclusions. Additionally, for antibodies that neutralized one but not all of the viruses tested, it can be assumed that the mAb was expressed. However, the conclusions in the results section should outline the positive neutralization results rather than determining the virus specificity of the mAbs since lack of virus neutralization may be related to insufficient quantity of antibody present in the supernatant.

We are looking forward to receiving your revised submission.

Reviewer Comments (if any, and for reference):

Reviewer's Responses to Questions

**Part I - Summary**

Reviewer #1: The authors have addressed all the concerns raised in their original manuscript. This paper now tells a logical and interesting story on an alternative method to discovery broadly neutralising antibody against DENV and ZIKV, as well as the under-appreciated role of IgA antibodies in the adaptive immune response against such flaviviral infections.

Reviewer #2: The authors have addressed all of my concerns except for their response to major concern #5

**Part II – Major Issues: Key Experiments Required for Acceptance**

Reviewer #1: None

Reviewer #2: In regards to my Major Concern #5

I do not find the authors argument compelling in the lack of standardization. I appreciate that purifying individual antibodies is time consuming, but it is also not challenging. That said, if the authors insist on using supernatants, they need to quantify the amount of antibody within that supernatant, which is a single and straightforward ELISA, and standardize their results based on this. I understand the point of trying to find neutralizing antibodies, but this approach and data presentation is not sound for publication and sets a bad precedent.

**Part III – Minor Issues: Editorial and Data Presentation Modifications**

Reviewer #1: None

Reviewer #2: (No Response)

PLOS authors have the option to publish the peer review history of their article (what does this mean?). If published, this will include your full peer review and any attached files.

Reviewer #1: No

Reviewer #2: No

Figure Files:

Data Requirements:

Reproducibility:

References:

---

## [Editor Report · Decision Letter 2]

28 Sep 2023

Dear Dr Goo,

We are pleased to inform you that your manuscript 'Single B cell transcriptomics identifies multiple isotypes of broadly neutralizing antibodies against flaviviruses' has been provisionally accepted for publication in PLOS Pathogens.

Best regards,

Julie Fox, Ph.D.

Academic Editor

PLOS Pathogens

Sonja Best

Section Editor

PLOS Pathogens

Kasturi Haldar

Editor-in-Chief

PLOS Pathogens

orcid.org/0000-0001-5065-158X

Michael Malim

Editor-in-Chief

PLOS Pathogens

orcid.org/0000-0002-7699-2064
---

## [Editor Report · Acceptance letter]

5 Oct 2023

Dear Dr Goo,

We are delighted to inform you that your manuscript, "Single B cell transcriptomics identifies multiple isotypes of broadly neutralizing antibodies against flaviviruses," has been formally accepted for publication in PLOS Pathogens.

Best regards,

Kasturi Haldar

Editor-in-Chief

PLOS Pathogens

orcid.org/0000-0001-5065-158X

Michael Malim

Editor-in-Chief

PLOS Pathogens

orcid.org/0000-0002-7699-2064